



# Evaluation of coupled and uncoupled ocean-ice-atmosphere simulations using icon_2024.07 and NEMOv4.2.0 for the EURO-CORDEX domain

Vera Maurer[1], Wibke Düsterhöft-Wriggers[2], Rebekka Beddig[2], Janna Meyer[2], Claudia Hinrichs[2], Ha Thi Minh Ho-Hagemann[3], Joanna Staneva[3], Birte-Marie Ehlers[2], and Frank Janssen[2]

[1]Deutscher Wetterdienst, Offenbach, Germany
[2]Bundesamt für Seeschifffahrt und Hydrographie, Hamburg/Rostock, Germany
[3]Institute of Coastal Research, Helmholtz-Zentrum Hereon, Geesthacht, Germany

**Correspondence:** Vera Maurer (vera.maurer@dwd.de)

**Abstract.** Evaluation results from the reanalysis-driven evaluation simulation for the years 1979–2021 with a regional coupled ocean–atmosphere model (ROAM) are presented. The coupled setup portrayed here is one of the first regional climate modeling systems to couple the ICON atmosphere model in climate limited-area mode (CLM) with the ocean model NEMO for the North and Baltic Sea (NBS), using a flux-based OASIS coupling approach. Along with the simulation with the coupled model

configuration ROAM-NBS, the simulations with the uncoupled components (ICON-CLM and NEMO-NBS, respectively) are analyzed and compared with various observational datasets. ROAM-NBS complements atmosphere-only climate projections with the same atmospheric model and setup, which will all be published in accordance with EURO-CORDEX specifications. Climate projections by ROAM-NBS will enrich the data available to support the German Strategy for Adaptation to Climate Change (DAS), especially for our target region, which are the German national waters.

In general, the mean model climate is well represented by all setups. The sea surface temperature bias is, on average, below $\pm 0.5\,\mathrm{K}$, with an improved representation of extreme sea surface height events in the uncoupled setup. SST biases of ROAM-NBS directly lead to biases of surface heat fluxes, wind speed, and precipitation over land. However, the mean influence on the land areas is negligible. The evaluations of ocean variables indicate a strong agreement of ROAM-NBS with NEMO-NBS. Compared to observations, both simulations overestimate sea ice concentration and extent in spring, especially in the Gulf

of Bothnia. Mean temperature profiles in the Baltic Sea show that both simulations generally reproduce observed profiles. ROAM-NBS exhibits a cold bias in deeper layers, especially in the Gotland Deep, while NEMO-NBS shows better agreement at the sea surface and bottom. Stratification analyses confirm a fresh bias in Baltic bottom layers, especially in the Gulf of Bothnia, with NEMO-NBS performing slightly better in this region. Major inflow events are captured but underestimated. Sea surface height and wind surge highly coincide with observational data, with NEMO-NBS slightly outperforming ROAM-NBS

in correlation. The marine heat wave (MHW) evaluation against observations in the North and Baltic Sea demonstrates that the simulations capture the inter-annual variability of MHW characteristics.

Overall, the coupled simulation demonstrates good performance for both the atmosphere and the ocean, and the setup is now ready to be used for producing coupled regional climate projections for Europe.



## 1 Introduction

In climate modeling and research, the results of regional coupled or ocean models are still underrepresented compared to global ones. While the regional downscaling of the atmosphere is coordinated through the CORDEX initiative (Giorgi et al., 2009), data from regional ocean models are still sparse and, due to the downscaling chain, can only be delivered with a considerable delay compared to the global climate simulations. Thus, one advantage of using a regional coupled ocean–atmosphere model for climate projections compared to the stand-alone ocean component is the independence of regional climate model (RCM)

projections from CMIP6. Moreover, the regional coupled model allows us to deliver consistent information on climate and climate change for the atmosphere and the ocean in the North and Baltic Sea (NBS) region, with a particular focus on the German coasts. For the Baltic Sea region, a number of investigations using regional coupled models were done within the framework of Baltic Earth (https://baltic.earth, last access: 9 July 2025). Gröger et al. (2021) review progress on coupled modeling in that context. The investigations considered in the review outline different aspects of the added value of regional coupled mod-

els. Gröger et al. (2021) summarize that only online coupled high-resolution ocean models can represent small-scale ocean processes accurately. They also conclude that the demonstration of the added value of coupled models over their uncoupled counterparts is often influenced by biases of datasets like runoff used for the forcing of the uncoupled versions. Christensen et al. (2022) analyzed RCM projections with and without ocean coupling, forced by global climate model (GCM) simulations provided with the fifth phase of the coupled model intercomparison project (CMIP5). Their focus was on climate change in

the Baltic Sea region. They showed that the coupled simulations can exhibit differences in future sea surface temperatures and sea-ice conditions compared to the respective uncoupled versions, which can locally modify the climate change signal. As shown by Gröger et al. (2021), different regional coupled ocean–atmosphere models have been in use for the NBS region. These are coupled versions of CCLM and NEMO (e.g. Pham et al., 2014; Primo et al., 2019; Ho-Hagemann et al., 2020), RCA4 and NEMO (Gröger et al., 2015; Dieterich et al., 2019), REMO and MPIOM (Sein et al., 2015) or Hirham and HBM

(Tian et al., 2013). Karsten et al. (2024) present a recent development of a coupled ocean–atmosphere model, which is coupling the atmosphere and the ocean component (CCLM and MOM5, respectively) using an exchange grid. However, the ocean domain of their coupled model only encompasses the Baltic Sea, which is a very small part of the whole EURO-CORDEX domain used for the atmosphere. A first version of a coupled regional ocean–atmosphere model incorporating ICON in climate limited-area mode (ICON-CLM) and NEMO was presented by Ho-Hagemann et al. (2024). The modeling system is called

GCOAST-AHOI, just as its earlier version (Ho-Hagemann et al., 2020). Ho-Hagemann et al. (2024) found that the new version of GCOAST-AHOI could well capture near-surface air temperature, precipitation, mean sea level pressure, and wind speed at a height of 10 m. However, there was a prevailing negative sea surface temperature (SST) bias of 1-2 K, which they attributed to an underestimation of the downward shortwave radiation at the surface.

Here, we introduce ROAM-NBS, a new version of a regional coupled ocean–ice-atmosphere modeling system covering the

full EURO-CORDEX domain for the atmosphere and the North and Baltic Sea for the ocean. ROAM-NBS combines the ICON-CLM atmosphere model (version icon_2024.07) with the NEMOv4.2.0 ocean model and the Sea Ice modelling Integrated Initiative (SI3) thermodynamic sea ice model, coupled via OASIS3-MCT using a flux-based exchange approach.



Compared to the version by Ho-Hagemann et al. (2024), ROAM-NBS is based on a later NEMO version and includes a new ocean bathymetry, which is particularly designed for a good representation of the German coastline. Moreover, our setup also integrates a refined treatment of radiation in NEMO based on prescribed chlorophyll distributions, supporting a realistic representation of shallow and stratified shelf seas. With the use of a later ICON release, a more recent NEMO version with major updates, higher-resolution coastal bathymetry, and enhanced representation of surface fluxes and radiation, ROAM-NBS represents a methodological advance over previous GCOAST configurations.

Using ROAM-NBS and different configurations of GCOAST-AHOI described by Ho-Hagemann et al. (2020, 2024), which all employ an online coupled ocean for the NBS region, CMIP6 climate projections will be downscaled for the EURO-CORDEX region (Jacob et al., 2014). These coupled regional climate projections will complement the RCM simulations of CMIP6-CORDEX (https://github.com/WCRP-CORDEX; last access: 23 May 2025). The simulation status is updated on a regular basis and can be viewed at https://wcrp-cordex.github.io/simulation-status/CORDEX_CMIP6_status.html#EUR-12 (last access: 23 May 2025). *The evaluation simulation analyzed in this article defines the setup of ROAM-NBS that will be used for the downscaling.* Additionally to publishing the data on the ESGF nodes within the EURO-CORDEX community, ROAM-NBS will be applied to generate an ensemble of climate projections that can be used for climate adaptation measures in German national waters. Related evaluations will be published on https://das.bsh.de (last access: 9 July 2025).

The comparison of coupled simulations against their uncoupled counterparts, which are forced by high-quality reanalyses like ERA5 (Hersbach et al., 2020) at the ocean–atmosphere interface, can never be a fair one. Thus, the most important added value of using a regional coupled model for climate projections is not shown when evaluating the reanalyses-driven evaluation simulation, where we can provide good forcing data for both components for uncoupled simulations. It is rather in the next step, when we are generating the coupled regional historical simulations (i.e., for the historical time period as the reanalysis-driven evaluation simulation, but with GCM data as lateral boundary conditions) and the climate projections, which are not part of this article. However, even if the added value of coupling cannot be fully quantified from reanalysis-driven simulations, the evaluation provides an essential test of consistency and robustness.

The main aim of this article is to analyze the general performance of the coupled evaluation simulation considering the atmosphere, the ocean, and the sea ice components. We evaluate whether the coupling leads to any significant degradation or drift in mean climate, and whether major climate-relevant features such as SSTs, near-surface air temperature, surface heat fluxes, stratification, and extreme sea level are consistently represented. Therefore, we are comparing the individual components against measurement data as well as the coupled simulation against the simulations of the standalone versions of the atmosphere and the ocean–sea-ice component. As each component can introduce new biases into the system, it has to be assured that the coupled system gives comparable results as the individual components.

After giving an overview on the coupled system and its individual components in Sect. 2, the evaluation is presented in two parts: In Sect. 3, the focus is on the evaluation of the mean climate. Surface variables are validated on the model domains of the atmospheric and ocean components, respectively. For profile data, the focus lies on evaluating the Baltic Sea and its complex stratification. Section 4 analyzes the ocean variability and selected extreme events such as Baltic inflow, storm surges, and marine heat waves. As the ocean component is of additional benefit compared to most other CMIP6-CORDEX simulations, a





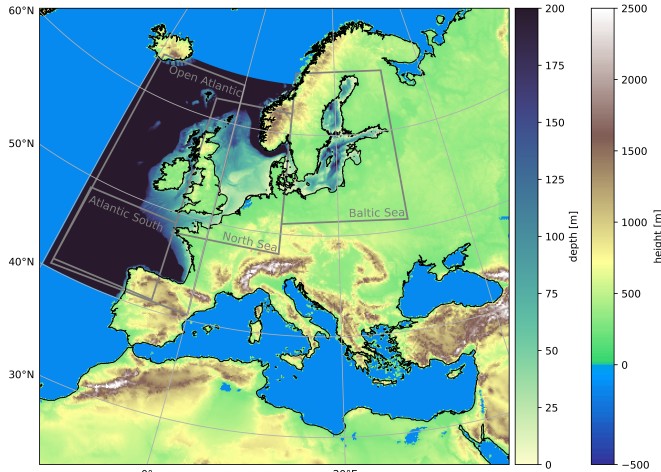

**Figure 1.** EURO-CORDEX domain (EUR-12) used for ICON-CLM (the corresponding colorbar is labeled with "height") and NEMO-NBS bathymetry (colorbar labeled with "depth"). Grey quadrangles denote boxes used in the evaluation.

particular focus is put on the evaluation of NEMO-NBS and the ocean part of ROAM-NBS. Main climate and extreme events are primarily assessed in German national waters as their severity and frequency in the presence and in the future are relevant

for the planning of climate change adaptation.

## 2 General model description and experiment setup

The ROAM-NBS coupled model comprises regional atmosphere, ice and ocean components adapted for high-resolution climate applications over Europe and its adjacent seas. Below, we describe the setup and configuration of each component, the coupling strategy, and the experiment design.

### 2.1 ICON-CLM

The initial version of ICON-CLM was set up by Pham et al. (2021) and was thereafter further developed by the Climate limited-area Modelling (CLM) community (https://www.clm-community.eu; last access: 15 July 2025) and within different projects funded by the German Federal Ministry of Education and Research. For the European domain (Fig. 1) with a horizontal resolution of about 12.1 km, the setup of ICON-CLM is largely based on the setup used for numerical weather prediction

(NWP) by the German Weather Service (DWD). An overview of the set of physical parameterizations used is given by Prill et al. (2024). Compared to the ICON-NWP configuration, ICON-CLM is adapted for climate applications: ICON-CLM uses temporally varying sea surface temperature and sea-ice concentration (*siconc*) fields prescribed from a coarser model or by reanalyses together with the lateral and upper boundary conditions. At the upper boundary, a nudging is applied using a nudging coefficient which is 1 at the model top of 23.5 km and <0.1 in the ninth model layer from top (14.06 km). To account for





temporally varying insolation, the CMIP6 forcing for spectral solar irradiance as provided by Matthes et al. (2017) is read in. Furthermore, the yearly varying greenhouse-gas forcing of CMIP6 is used.

To make ICON more suitable for climate projections, the model had to be adapted for the usage of the transient climatologies of aerosol and ozone. These adaptations were mainly done for the new global ICON-based Earth System Model, ICON-XPP (Müller et al., 2025a, b): For ozone, the CMIP6 dataset (Checa-Garcia, 2018) is used. The aerosol climatologies used originate from ECHAM6 (Stevens et al., 2013), namely the anthropogenic aerosol climatology MACv2 developed by Kinne et al. (2013) and Kinne (2019), but in the simple-plume version (MACv2-SP, Stevens et al., 2017), and an extended volcanic aerosol climatology based on Stenchikov et al. (1998). However, the Kinne aerosols provide different input parameters compared to the default aerosol climatology used in ICON for NWP. While different species are available from the aerosol climatology of Tegen et al. (1997) in the NWP version, which allows to infer properties like hygroscopicity, this distinction of different species is not available for the Kinne aerosols. In NWP, the aerosol properties are used to determine the influence of aerosols on cloud droplet numbers, which are an important quantity in cloud microphysics, with a secondary effect on radiation. Using the Kinne aerosols, the aerosol-cloud coupling was lost and had to be replaced with an alternative treatment. It was decided not to use the scaling factors available from the simple plume scheme (Fiedler et al., 2019), as these depend on assumptions, but to use a climatology of cloud droplet number concentrations (CDNCs) derived from satellite measurements. The non-transient, but monthly varying climatology used here is from Gryspeerdt et al. (2022) and was adjusted with a climatology of ECMWF (MODIS CDNC climatology based on Bennartz and Rausch (2017) and Grosvenor et al. (2018)).

Another difference to the NWP setup is the use of a soil-type distribution of the Harmonized World Soil Database v2.0 provided by the Food and Agriculture Organization of the United Nations (FAO). As recently in NWP, the urban parameterization TERRA_URB was switched on (Zängl et al., 2025).

Note that the simulations and the raw output of ICON are done on an icosahedral, irregular grid (R13B5 in ICON notation). However, the raw output is interpolated using the nearest-neighbour approach onto the common EUR-12 grid (Fig. 1) and all evaluations for the atmospheric part are, therefore, also done on this rotated longitude-latitude grid.

For EURO-CORDEX, the evaluation simulation for ICON-CLM that will be published on ESGF was generated within the project "Updating the data basis for adaptation to climate change in Germany" (UDAG). Hereafter, this simulation will be called ICON-CLM-UDAG. The latest tuning for ICON-CLM was done in a common effort of the UDAG project and the CLM community (Geyer et al., 2025) and is also applied in our ICON-CLM and ROAM-NBS simulations apart from one namelist setting, which will be discussed in Sect. 3.2.1.

For ROAM-NBS as well as for UDAG, the open source release icon-2024.07 was used. However, the OASIS3-MCT interfaces necessary for the NEMO coupling are not included in the official version as YAC is the only coupler supported by the ICON consortium. Therefore, the modifications of the feature branch containing the OASIS interfaces are made available via github.



## 2.2 NEMO-NBS

The ocean component is based on the Geesthacht Coupled cOAstal model SysTem (GCOAST) setup, developed within the coastal model framework of HEREON (Ho-Hagemann et al., 2020; Grayek et al., 2023; Staneva et al., 2016). In the original

GCOAST setup, NEMO3.6 is used in the regional setup for the Northwest Shelf and the Baltic Sea ((Staneva et al., 2016; Grayek et al., 2023). The main feature of the GCOAST setup is the enhanced representation of the coastal processes in the larger European shelf domain (Fig. 1), together with North and Baltic Seas, allowing a lateral resolution of 2 nm and mixed $\sigma$-$z^*$-coordinates with 50 vertical layers. This vertical coordinate allows the representation of the deeper and shallower regions at the same time. The physical parameterizations and their settings were chosen therefore chosen as in (Ho-Hagemann et al.,

2020).

Some changes were made to the original GCOAST version: In the current work, the updated NEMOv4.2.0 (Gurvan et al., 2022) with a new sea ice model SI3 (Vancoppenolle et al., 2023) instead of NEMOv3.6 was used. Besides the model version, an updated bathymetry from the European Marine Observation and Data Network (EMODnet) framework (EMODnet Bathymetry Consortium, 2020), which comes with finer resolution, especially relevant for the coastal representation, substi-

155 tuted the General Bathymetric Chart of the Oceans (GEBCO)-based bathymetry. Further, manual fine-tuning of the German coasts and Danish straits and a Laplacian smoothing were applied to the EMODnet data. In Fig. 1, the depth deeper than 200 m is set to dark blue to accentuate the European North West Shelf area. Another modification to the original GCOAST setup was made in the radiation scheme: a three-band RGB radiation scheme instead of a two-band scheme was used. This scheme allows for a differentiated radiation treatment of the North and Baltic Seas, enabling better representation of highly stratified zones.

The spatial variability in radiation is represented through a two-dimensional climatological field, which captures the mean chlorophyll concentration. This field serves as a prescribed parameter for the RGB radiation scheme. Since neither ROAM-NBS nor NEMO-NBS include a biogeochemical module, the chlorophyll field is derived from literature and observational datasets: Schernewski et al. (2006) for the Baltic Sea, OSPAR Comission (2017) for the North Sea, and NASA Earth Observations (2024) for the Atlantic Ocean. Due to this change, a different radiative penetration of the sea surfaces of the shallower

Baltic Sea and the deeper North Sea could be achieved. The state-of-the-art Thermodynamic Equation of Seawater (TEOS-10, https://www.teos-10.org, last access: 26 June 2025) is applied within NEMO-NBS for the calculation of state variables.

The river runoff dataset was provided by the German Bundesanstalt für Gewässerkunde and combines observational data in German national waters with model results of the WaterGAP hydrological model (Müller Schmied et al., 2021).

In the NEMO stand-alone setup, atmospheric forcing fields of the ERA5 reanalysis (Hersbach et al., 2020) in an hourly tem-

170 poral resolution are used. These comprise wind velocities at 10-m height, air temperature and dew point temperature at 2-m height, mean sea level pressure, downward solar and thermal radiations. The surface turbulence and momentum fluxes are estimated using the ECMWF (2018) bulk formulation as implemented in the Aereobulk package (Brodeau et al., 2017). An additional influence is set by adding an atmospheric pressure as inverse barometer sea surface height to the ocean momentum equation.

The main tidal constituents M2, N2, 2N2, S2, K2, K1, O1, Q1, P1, and M4 for both NEMO-NBS as well as the coupled



ROAM-NBS are provided at the lateral boundaries using the FES2014 data set (Lyard et al., 2021).

Within NEMO-NBS and ROAM-NBS, only the ice thermodynamics of the SI3 sea ice model is applied, using five ice categories and two ice layers.

This setup represents one of the first implementations of NEMOv4.2.0 with SI3 in a fully coupled regional system for the European shelf, North and Baltic Sea with direct applicability to coastal hazard and climate risk assessments.

### 2.3 Coupling via OASIS3-MCT

ICON and NEMO are coupled via the OASIS3-MCT_5.0 coupler (Craig et al., 2017). Hereafter, we will refer to OASIS only for the sake of brevity. The interfaces within the ICON code are based on the implementation described by Ho-Hagemann et al. (2024). On the NEMO side, the OASIS interfaces are used as provided with NEMOv4.2.0. In our setup, we are using exclusively the approach of the flux coupling (Ho-Hagemann et al., 2024), which has the advantage over the bulk coupling that both the atmosphere and the ocean model see the same turbulent fluxes. Additionally, as ICON incorporates a tile approach, the flux coupling ensures the best possible local conservation of energy on non-common grids without using an exchange grid as, for example, Karsten et al. (2024); when variables are sent from the atmosphere at lower horizontal resolution to the ocean via OASIS, tiled quantities are used. This approach is very similar to the one used in ICON-XPP (Müller et al., 2025b). These tiled quantities are representative for the ocean and sea ice fractions, respectively, in each atmospheric model grid cell, and are used by the ocean for the respective computations for open ocean and sea ice. The respective quantities are solar shortwave radiation, momentum flux, and the so-called non-solar radiation, which is the sum of sensible heat flux, latent heat flux, and long-wave radiation. As the ice fraction is sent from the ocean model to the atmospheric one via OASIS, the ice fractions of both models are consistent. In the default NWP and ICON-CLM setup, the sea-ice scheme calculates the heat transfer within the sea ice, dependent on the current ice depth and a standard bottom temperature. It cannot generate new sea ice, but the ice thickness can decrease and ice albedo and surface temperature are determined depending on the heat transfer as well as on snow lying on the ice. To make the thermal part more consistent with the NEMO/SI$^3$, which has a more detailed treatment of thermodynamical processes within sea ice, the sea-ice albedo, thickness and surface temperature are transferred from the ocean to the atmosphere in our setup. The ocean albedo over water is not sent, as this option is not available in the default coupling setup of NEMOv4.2.0. Only the one over ice is available. The complete list of variables exchanged during the coupling in ROAM-NBS is given in Table 1. For selected variables, a global conservation is applied, which means that the area average is compared before and after horizontal interpolation. By selecting "GLBPOS" in the OASIS settings, the difference between the area average on the target minus the source grid is distributed proportionally to the value of the original field.

Because only the respective portions of fluxes for the ocean and sea-ice tiles are used, flux contributions from land points are completely excluded when fluxes are sent from the atmosphere to the ocean, which also ensures a consistent exchange of energy and momentum along the coastlines. As suggested by Mechoso et al. (2021) for models that incorporate the tile approach, the ocean fraction of the atmospheric model is adapted to that of the ocean model, which is obtained by interpolating the binary land-sea mask of the ocean model to the atmospheric grid. The land and lake fractions of the atmospheric model are adapted accordingly. For a correct treatment of coastal points, great care was put on identical interpolation methods for the





**Table 1.** Variables exchanged between the atmosphere and the ocean via OASIS; for variables denoted by [*], global conservation is applied by OASIS after horizontal interpolation; NEMO variables denoted by [**] are aggregated over all ice categories before sent to the atmosphere; NEMO variables denoted by [***] are rotated from the geographical to the local grid and staggered onto the U,V grid within NEMO; "ocean" or "ice" are given in parentheses for ICON variables to indicate that only the part of the variable on the respective tile is used.

| variable | NEMO variable | ICON variable |
|---|---|---|
| ocean → atmosphere | | |
| SST | ts (as potential temperature) | t_seasfc ; t_s ; t_s_t(ocean) |
| sea-ice fraction | fr_i | fr_seaice |
| sea-ice albedo | alb_ice[**] | albdif_t |
| sea-ice thickness | h_i[**] | h_ice |
| sea-ice (surface) temperature | tn_ice[**] | t_ice |
| atmosphere → ocean | | |
| solar radiation[*], ocean tile | qsr | swflxsfc_t(ocean) |
| solar radiation, ice tile | qsr_ice | swflxsfc_t(ice) |
| non-solar radiation[*], ocean tile | qns | lwflxsfc_t(ocean) + lhfl_s_t(ocean) + shfl_s_t(ocean) |
| non-solar radiation, ice tile | qns_ice | lwflxsfc_t(ice) + lhfl_s_t(ice) + shfl_s_t(ice) |
| u-momentum flux, ocean tile | utau[***] | umfl_s_t(ocean) |
| u-momentum flux, ice tile | p_taui[***] | umfl_s_t(ice) |
| v-momentum flux, ocean tile | vtau[***] | vmfl_s_t(ocean) |
| v-momentum flux, ice tile | p_tauj[***] | vmfl_s_t(ice) |
| precipitation[*], liquid part | $\mathrm{zemp} = \mathrm{evap} - (\mathbf{rain} + \mathrm{snow})$ | rain_con_rate + rain_gsp_rate |
| precipitation[*], solid part | $\mathrm{zemp} = \mathrm{evap} - (\mathrm{rain} + \mathbf{snow})$ | snow_con_rate + snow_gsp_rate |
| evapotranspiration[*] | $\mathrm{zemp} = \mathbf{evap} - (\mathrm{rain} + \mathrm{snow})$ | qhfl_s |
| sublimation | evap_ice | qhfl_s_t(ice) |
| mean sea-level pressure | $\mathrm{apr}$ ; $\mathrm{ssh\_ib} = -(\mathbf{apr} - \mathrm{rpref})/(\rho * g)$ | pres_msl |

NEMO land-sea mask and the NEMO fields to the ICON grid during coupling, as well as on the consistency of the common land-sea mask and the masks used by OASIS. Thus, the flux coupling as included in ROAM-NBS is a good compromise between simplicity and correctness.

## 2.4 Overview of experiments

The evaluation simulation was run for the coupled model (ROAM-NBS) and for both the atmospheric and the ocean compo-
nents individually (ICON-CLM and NEMO-NBS, respectively), i.e., we are overall evaluating and comparing three simulations for the years 1979–2020. In Sect. 3.2.1, a short comparison to the evaluation simulation from the UDAG project (called ICON-





CLM-UDAG) is shown additionally. We started our simulations in September 1978, but the first 4 months were disregarded from the evaluation. In some parts of the evaluation, shorter periods were chosen due to limited data availability. Only for the evaluation of marine heat waves, also the last year of the simulation (2021) is included. The experiments were started in

September 1978 because of the stable thermal mixed layer in the ocean. Starting in January would mean starting in the middle of the winter season when thermal conditions are unstable and the sea ice is already evolved, which proved disadvantageous for the model initialization, especially in coupled mode. To avoid a so-called "cold start", a spin-up for the ocean component was carried out with NEMO-NBS for the years 1974–1978, starting with an initial field for the temperature and the salinity. This initial field was, in turn, a combined product from the ORAS5 reanalysis and the Baltic reanalysis product provided by

SMHI (personal communication) based on Hordoir et al. (2019).

For a better comparison of NEMO-NBS and the ocean part of ROAM-NBS, it was decided to use the same observation-based dataset for the runoff for both experiments. ICON-CLM as well as the atmospheric part of ROAM-NBS were started from the ERA5-driven ICON-CLM-UDAG simulation, which had been running since 1940.

For the atmospheric part, ERA5 reanalyses were used as lateral boundary conditions as prescribed by the CORDEX-CMIP6

experiment protocol (CORDEX, 2025). They are available at 3-hourly resolution and they are also used to prescribe SST and sea ice in the uncoupled ICON-CLM simulation. Also for ROAM-NBS, ERA5 SST and sea ice fields are used in ocean regions outside of the NEMO-NBS domain. As lateral boundary conditions for the ocean, the temperature, salinity, zonal and meridional ocean velocities and sea surface height (SSH) fields from the ORAS5 global reanalysis data set (Zuo et al., 2019) are used. To ensure the model zero point to be consistent with the one from ORAS5, the ORAS5 mean SSH at the boundaries

is calibrated to the SSH at Helgoland. The prescribed fields were used at a monthly resolution as daily data were not available for the whole simulation period.

## 3    Evaluation of the mean climate

To assess the realism of the coupled ROAM-NBS system, we evaluate its representation of the present-day climate over the North and Baltic Seas in comparison with the atmosphere-only (ICON-CLM) and ocean-only (NEMO-NBS) simulations. The

seasonal means are calculated for the main ocean-atmosphere surface variables to evaluate the performance of ROAM-NBS and its individual components with respect to climate time scales. Mean temperature and salinity profiles are validated in the Baltic to assess the ocean components' ability to model highly stratified regions. Statistical values for the sea surface height are presented for stations throughout the ocean domain.

### 3.1    Sea surface temperature

The most crucial variable at the interface of the atmosphere and the ocean is the sea surface temperature as the first indicator for the models' performance and the proper coupling. Here, the simulated sea surface temperature is compared to satellite observations (Copernicus ESA SST CCI and C3S reprocessed sea surface temperature analyses, https://doi.org/10.48670/moi-00169). Detailed information on the quality of the observation data is given in the quality information document of the dataset that can





(a) ROAM-NBS vs. OBS  (b) ROAM-NBS vs. NEMO-NBS

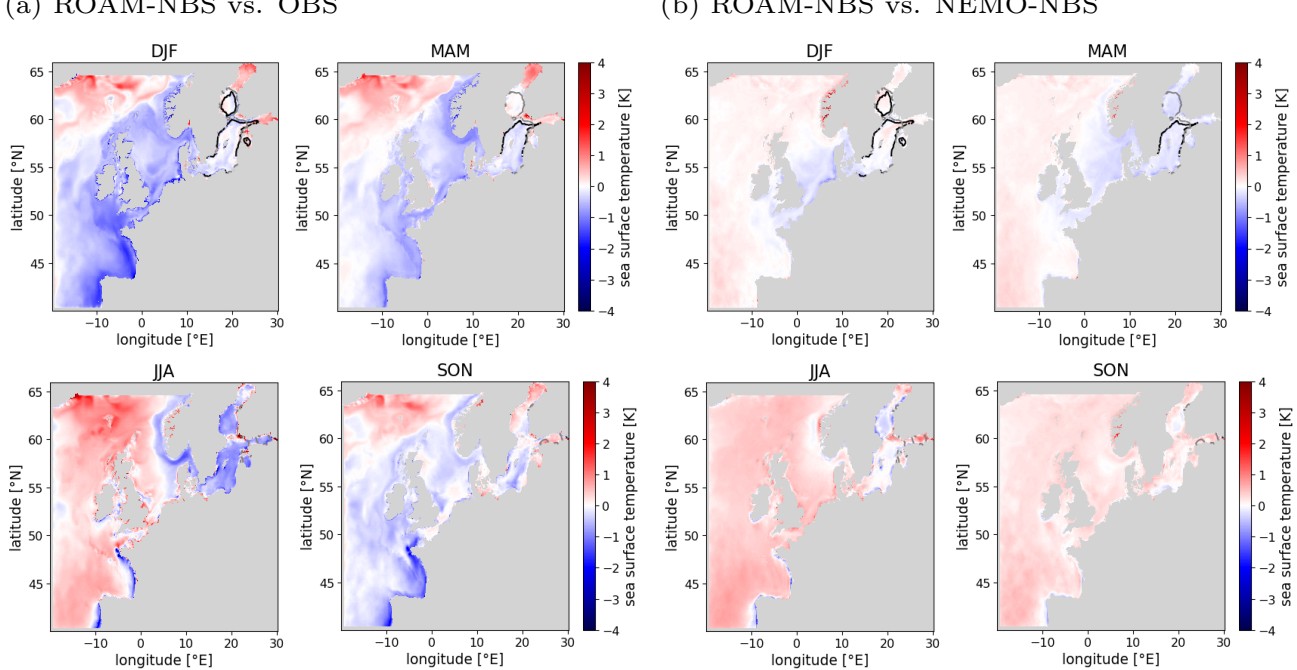

**Figure 2.** Seasonal mean sea surface temperature difference between ROAM-NBS and Copernicus observations (a) and ROAM-NBS and NEMO-NBS (b) for the period (September 1981–November 2020) with ice contours for Copernicus observations (grey) and model data (black).

also be found on the product-specific Copernicus website. Figure 2a shows the difference of the simulated seasonal mean sea

surface temperature (ROAM-NBS) to observations. The observed maximum sea ice extent (mean sea ice concentration > 0.15) during the winter and spring seasons is indicated by grey contours to highlight the regions where SSTs may not be analyzed. The black contours indicate the simulated ice contours. ROAM-NBS shows a cold bias of locally up to 1.5 K in the shallower regions of the Northwest European shelf and in the Biscaya region during the colder seasons (SON, DJF, MAM). In the Baltic, it is around zero in these seasons. In summer, a positive bias can be observed in most parts of the NBS domain apart from the

Baltic where the bias becomes negative (up to −1 K in some years, cf. Fig. 3d). A persistent warm bias pattern is found near the northern Atlantic boundary east of Iceland. This temperature bias could originate in the northward heat transport by the Gulf Stream at the surface and a too weak vertical diffusion/downward transport of heat at the model's boundary. The border between the positive and negative differences in all seasons except summer seems to envelope the northwest European shelf with a stronger bathymetry gradient, which could be a hint for an underestimation of Atlantic on-shelf transport (Ricker and Stanev,

2020) into the North Sea. In Fig. 2b, the difference of the seasonal mean sea surface temperature of ROAM-NBS to NEMO-NBS is shown. The coupled simulation is generally slightly warmer than the uncoupled one, especially in summer (JJA). For ICON-CLM as well as for ROAM-NBS, an important tuning target was the reduction of the positive surface shortwave radi-





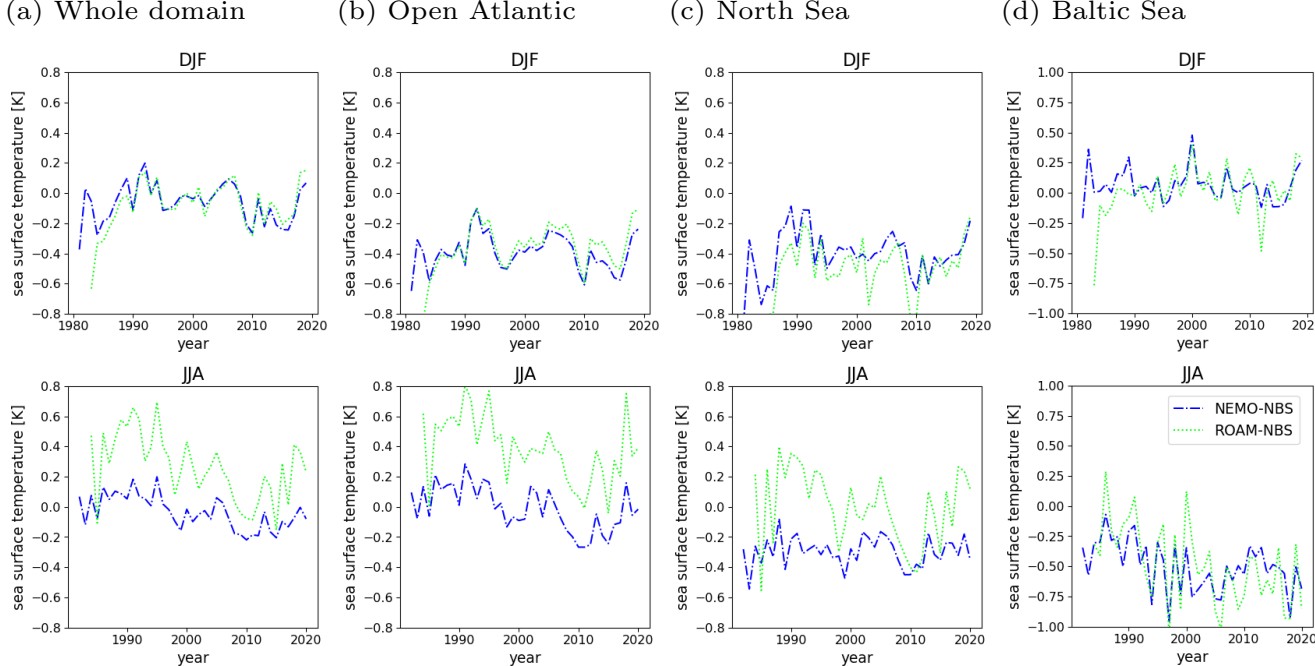

**Figure 3.** Bias of winter (DJF) and summer (JJA) seasonal mean SST time series for NEMO-NBS and ROAM-NBS to Copernicus observation data for the whole NEMO-NBS domain (a), Open Atlantic (b), the North Sea (c) and the Baltic Sea (d).

ation bias over land and ocean. In the current version, a tuning of the cloud cover is applied, which decreases the bias to a minimum, so that it is lying between $\pm 10\,\mathrm{W\,m^{-2}}$ (Fig. A1) in the NBS region compared to CERES (NASA/LARC/SD/ASDC, 265 2019). This reduction of the radiation bias contributed to a decrease of the positive SST bias in summer. However, it was not possible to reduce it further by a tuning of the atmospheric part. Eddy diffusivity in the eastern Atlantic was parameterized to be an order of magnitude higher than in the western Atlantic, which may have contributed to the cold bias observed near the French and Portuguese coasts. The time series of the seasonal SST biases against the Copernicus observations are shown for selected regions in Fig. 3. The locations of these regions (Baltic Sea, North Sea, Open Atlantic, and the whole domain) can 270 be seen in Fig. 1. The SST bias does not increase with the evolving simulation (Fig. 3) except for the spring season (MAM) in the Baltic Sea where the SST bias decreases within the period of 1981–2020 (cf. Fig. A4). For the whole domain, the time series shows a maximum bias of $\pm 0.5\,\mathrm{K}$ for NEMO-NBS (Fig. 3a). In winter (DJF) and spring (MAM), the area-averaged SST bias is very similar for ROAM-NBS. In summer (JJA) and autumn (SON), the SST bias of ROAM-NBS is about 0.4 K higher for ROAM-NBS than for NEMO-NBS. The time series for the absolute area-averaged SSTs demonstrate that the year-to-year 275 variability as well as warming trends in the North and Baltic Sea are well reproduced by both simulations (cf. supplementary Fig. A4).





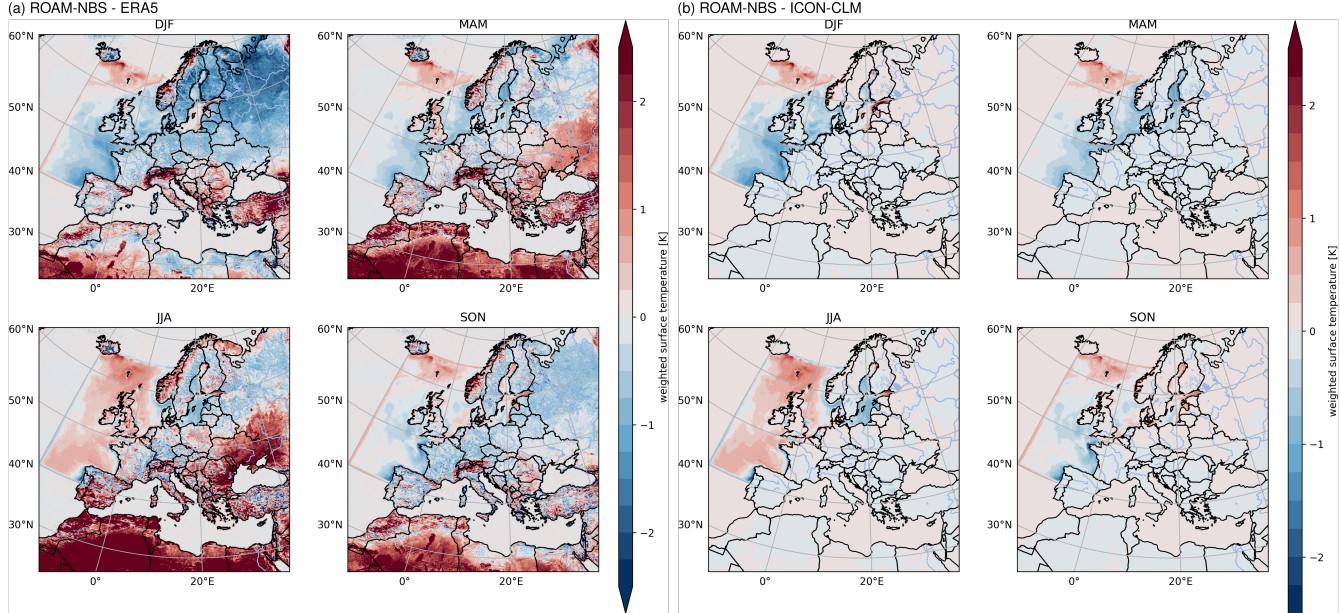

**Figure 4.** Mean seasonal surface temperature difference between ROAM-NBS and ERA5 (a) and ROAM-NBS and ICON-CLM (b) for the whole evaluation period (1979–2020).

## 3.2 Meteorological conditions

### 3.2.1 Surface and near-surface temperatures

Over the ocean, the seasonal mean surface temperature differences between ROAM-NBS and ERA5 (Fig. 4 a) mainly reflect the bias of NEMO against Copernicus observations with too low SSTs in winter and too high SSTs in summer. The differences between ROAM-NBS and ICON-CLM (Fig. 4 b) over the ocean are identical to the ROAM-NBS/ERA5 differences by design, as ICON-CLM is using ERA5 SSTs as a lower boundary condition over the ocean. Over land, the differences are small (i.e. $< \pm 0.25$ K) compared to the biases against ERA5.

For our ICON-CLM simulation, identical parameter settings as in ICON-CLM-UDAG were used apart from the minimum diffusion coefficient for heat, which was increased in winter (DJF). The higher minimum diffusion coefficient increases vertical mixing especially in stable boundary layers, causing downward mixing of warmer air and an increase of near-surface temperatures. This adaptation was done to compensate for the wintertime SST cold bias of NEMO-NBS that can also influence the air temperatures directly downstream of the ocean regions. However, from the evaluation of air temperature at 2 m (tas for "surface air temperature") against the E-OBS dataset (Cornes et al., 2018), which is available over land only, it is obvious that too low diurnal maximum temperatures (tasmax) are still prevailing (Figs. 5c and A2b) in ROAM-NBS. At the same time, minimum temperatures (tasmin) have a small warm bias compared to E-OBS (Figs. 5b and A2a). For the diurnal average tas, the ICON-CLM-UDAG simulation is by up to 0.7 K too cool in winter for the whole E-OBS domain (Fig. 5a). The diurnal




minimum (tasmin) in ICON-CLM-UDAG is too high in almost all regions and all months, with maximum biases of about 0.45 K in summer for the whole E-OBS domain average (Fig. 5b). At the same time, the diurnal maximum is up to 1.2 K too low in ICON-CLM-UDAG (Fig. 5c). Thus, the amplitude of the diurnal cycle is too low on average.

The increased minimum diffusion in ICON-CLM decreases the diurnal mean temperature bias in winter (DJF), while it is slightly increased in May to September. Accordingly, minimum and maximum temperatures are increased, which means that the already positive minimum temperature bias is getting larger. The largest positive minimum temperature bias of 0.8 K can be seen in February in Scandinavia.

To sum it up, the wintertime cold bias of ICON-CLM-UDAG is reduced by the adapted tuning parameter in ICON-CLM by about 0.2 K, at the cost of an increased warm bias of the diurnal minimum by about the same order of magnitude. However, the absolute values of the negative tasmax bias are still larger than the positive tasmin bias apart from July and August.

Comparing ROAM-NBS and ICON-CLM, we can say that the positive diurnal mean and minimum bias are slightly higher in ROAM-NBS than in ICON-CLM, with a very similar bias for the diurnal maximum. As expected, the wintertime SST cold bias of the ocean is also slightly reflected in the temperatures over land.

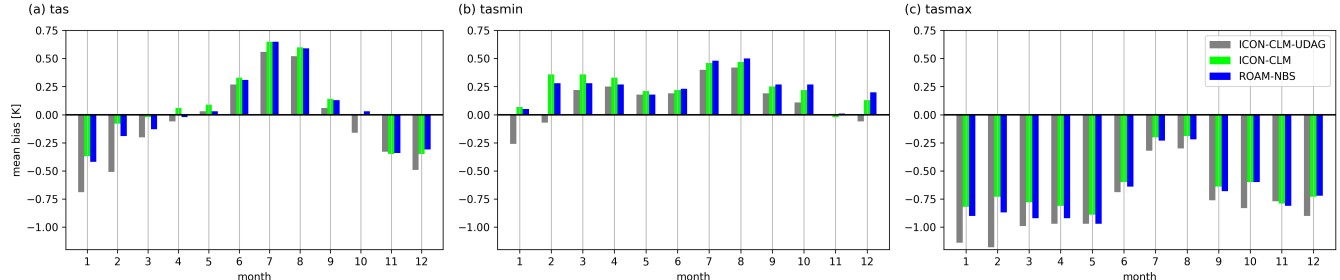

**Figure 5.** Monthly and spatial mean biases of tas, tasmin, and tasmax against E-OBS for ICON-CLM-UDAG, ICON-CLM, and ROAM-NBS, averaged over 1979–2020.

### 3.2.2 Precipitation and flux differences

For precipitation, the differences between ROAM-NBS and ICON-CLM reflect the differences in evaporation or latent heat flux (hfls) over the ocean (Fig. 6): Precipitation over the ocean is higher by up to $0.15\,\mathrm{mm\,day^{-1}}$ in ROAM-NBS in regions where evaporation is higher, as for example over the Atlantic ocean and North Sea in summer or the Baltic Sea in winter. Vice versa, precipitation is less by up to $-0.15\,\mathrm{mm\,day^{-1}}$ when evaporation is less, like over the Atlantic ocean and the North Sea in summer. However, these differences are small compared to the absolute precipitation biases over land, which exceeds $\pm 0.6\,\mathrm{mm\,day^{-1}}$ in various regions (Fig. A1b).

The mean values of precipitation and evaporation differences between ROAM-NBS and ICON-CLM are also given for different ocean domains in Fig. 7, together with the differences for sensible heat flux (hfss), 10 m wind speed (sfcWind) and skin




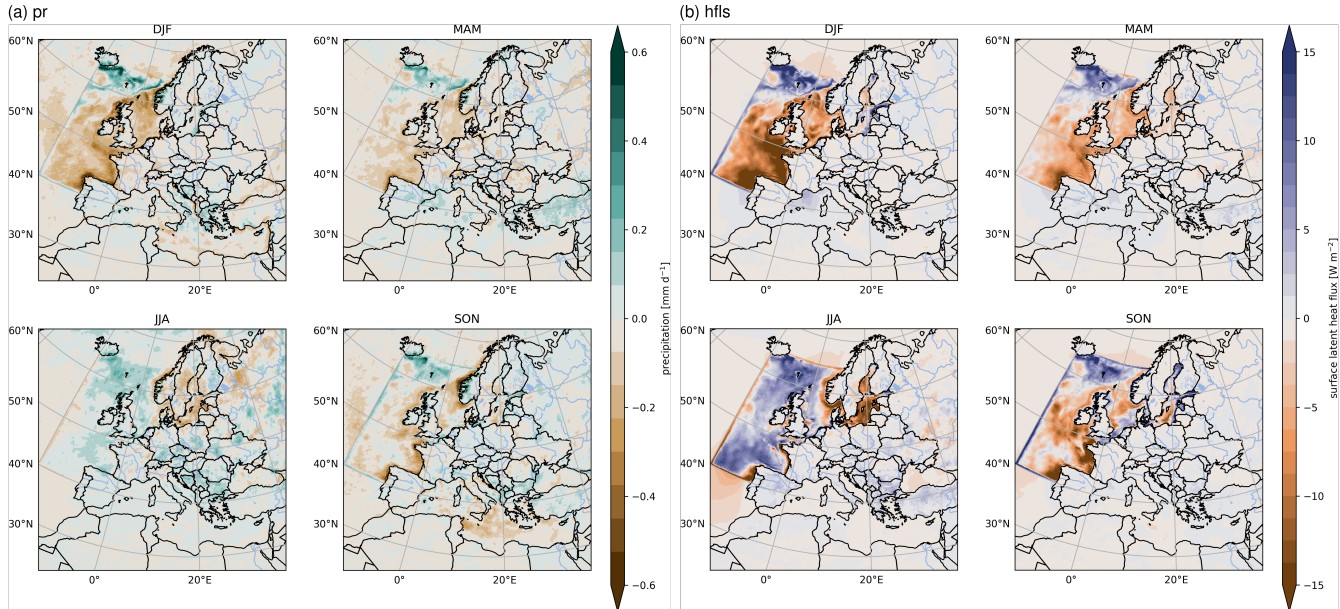

**Figure 6.** easonal mean difference of precipitation (pr, a) and latent heat flux (hfls, b) of ROAM-NBS against ICON-CLM, averaged over 1979-2020.

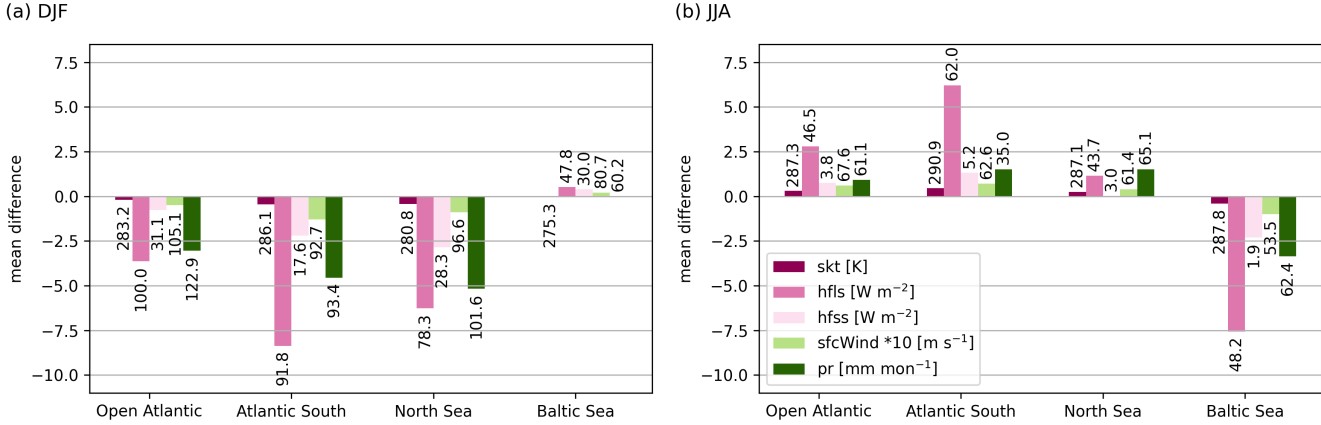

**Figure 7.** Mean absolute differences of ROAM-NBS against ICON-CLM for winter (a) and summer (b) for different quantities (see legend), averaged over four different regions, excluding the land parts, and over 1979-2020. The numbers indicate the mean values of ICON-CLM for the respective quantities.





temperature (skt). The absolute differences show clearly that the sign of the skt difference determines the sign of the flux, sfcWind and pr differences. For example, for a lower skt of ROAM-NBS as in the Open Atlantic, Atlantic South and North Sea boxes in DJF and the Baltic Sea box in JJA, also hfls, hfss, sfcWind and pr are lower than in ICON-CLM. Accordingly, all differences in the Open Atlantic, Atlantic South and North Sea boxes in JJA are positive. The only exception is the Baltic Sea

in DJF, where the spatially averaged skt differences is very small (below $-0.05$ K). The absolute area-averages for ICON-CLM (given as numbers in Fig. 7) show that the fluxes and therefore, also the flux differences become higher over a warmer ocean surface. Looking at the Atlantic South and the North Sea in DJF, for example, the skt difference is very similar in both regions ($-0.45$ K and $-0.43$ K, respectively), but the heat flux difference is larger for the Atlantic South ($-10.57$ W m$^{-2}$ for the sum of hfls and hfss compared to $-9.13$ W m$^{-2}$ for the North Sea). But the Atlantic South is, on average, warmer than the North Sea

(286.1 K compared to 280.8 K). However, the absolute temperatures of the ocean surface cannot explain all differences, otherwise the fluxes would be much higher in JJA than in DJF. The temperature and humidity of the overlying air masses as well as the mean wind speed also have an influence. The sfcWind differences are shown as their interpretation is more straightforward than for the momentum flux, which is given as u and v components. The sign of the sfcWind differences being in agreement with the flux differences shows that more mixing due to higher surface temperatures does not only result in stronger turbulent

heat fluxes, but also in more downward mixing of horizontal momentum and, thus, in higher near-surface wind speeds (and vice versa). In conclusion, SST biases, which are already present in NEMO-NBS and introduced into the coupled simulation, are directly reflected by the turbulent heat fluxes, near-surface wind speed and precipitation. However, these biases have no systematic impact on the land areas.

### 3.2.3 Wind speed


For a more detailed evaluation of simulated wind speed, DWD station observations along the German coast as well as the wind measurements at 100 m height at the FINO1 platform were used. The locations of the stations as well as of FINO1 are indicated in Fig. 8a. The surface station evaluation was done for hourly wind speed for the years 2011-2020, which was the period with the most common observations. FINO1 was evaluated only up to 2010, as the wake effect of the wind farms at

this location is too large after 2010 (e.g. Spangehl et al., 2023). For the surface stations, the nine closest grid points were used to capture the variability due to a potential mismatch between the coastline in the model and in the real world as well as unresolved geographical features. For FINO1, however, only the closest grid point was selected as the platform is located in the North Sea about 45 km away from the German coast and the measurement is taken at 100 m height, where surface effects have less influence. For both the surface station as well as the FINO1 comparison (Figs. 8b and c), an underestimation of wind

speed over the ocean by both ROAM-NBS and ICON-CLM can be seen. The violinplot for FINO1 illustrates that both the mean and the maximum wind speed are underestimated, as both the median and the maxima are smaller in the simulations. The underestimation seems to be strongest for stations further away from the mainland like those on lighthouses and floating devices ("UFS Deutsche Bucht", "Leuchtturm Kiel", "UFS TW-Ems", "Leuchtturm Alte Weser"). For most of these, the spatial variability is small (indicated by the dots lying closer together in Fig. 8b). However, at least the wind measurements on



(a)

(b)

(c)

**Figure 8.** Map with an overview of the evaluated station observations (a); mean station observations of hourly wind speed for 2011-2020, mean values and root-mean squared errors (rmse) of 9 closest grid points for each station for ROAM-NBS and ICON-CLM (b); violinplot displaying median, minimum and maximum values of hourly wind data at 100 m height at FINO1 and at the closest grid points in ROAM-NBS and ICON-CLM, for the years 2004-2010 (c).





(a) ROAM-NBS vs. OBS  (b) ROAM-NBS vs. NEMO-NBS

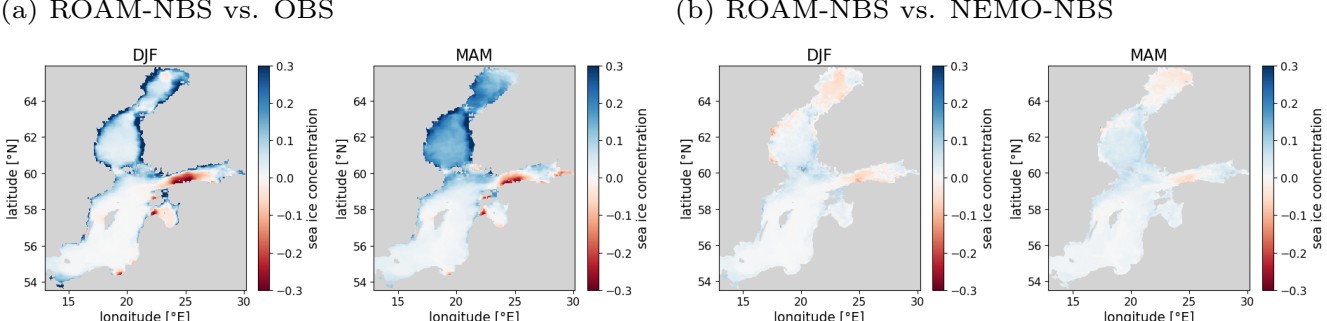

**Figure 9.** Differences in the mean sea-ice concentration for ROAM-NBS vs. Copernicus observations (a) and ROAM-NBS vs. NEMO-NBS (b) for winter and spring for 09/1981-11/2020.

lighthouses are not taken at a height of 10 m but further up at about 40 m, which explains the strong underestimation displayed in Fig. 8b. In most cases, ICON-CLM displays are very slightly higher wind speed than ROAM-NBS, but this is not surprising as the SST bias in ROAM-NBS is on average slightly negative which also causes a negative wind speed bias.

The underestimation of wind speed is weaker, but still discernible for stations on smaller islands like Helgoland, Büsum, Spiekeroog or Hallig Hooge. An exception is Norderney, but here it is unclear why the mean observed wind speed is by about

2 m s$^{-1}$ smaller than at Spiekeroog, which is relatively close. In contrast, for most stations *on* the coast or for Fehmarn, which is a larger island compared to those in the North Sea, the mean wind speed is well represented by both ROAM-NBS and ICON-CLM and also the RMSE values are small compared to other stations.

## 3.3 Oceanographic Conditions

### 3.3.1 Sea-ice concentration and extent

Figure 9 shows the differences in the seasonal mean sea ice concentration of NEMO-NBS and ROAM-NBS compared to Copernicus observations for winter and spring. Both models tend to overestimate the sea ice concentration, especially in MAM, i.e. towards the end of the sea ice season. The grey (observation) and black (model) contours representing the sea ice extent in Fig. 2 show a good agreement for both ROAM-NBS and NEMO-NBS against observational data for the winter season DJF. Nevertheless, in the spring season MAM, the amplification of sea ice extent is prominent for the Gulf of Bothnia. The

overestimation of sea ice concentration as well as extent in MAM might be due to a negative bias in the surface air temperature over the Bothnian Bay and Sea in winter and spring (Fig. 4) and an underestimation of salinity, especially in the Gulf of Bothnia (see the next subsection). Further, in Nie et al. (2023), the NEMO-SI3 model for version 4.0 was found to be most sensitive to the ice–ocean and air–ice drag coefficients and therefore to ice dynamics. Since only thermodynamical processes are modelled within ROAM-NBS and NEMO-NBS, further enhancements in results could be obtained by also considering ice dynamics or

further parameter tuning in the applied thermodynamic ice model.





### 3.3.2 Temperature

In addition to the sea surface temperature evaluation in Sect. 3.1, the evolution of temperature over time and mean temperature profiles are compared against observational profiles for the ROAM-NBS and NEMO-NBS simulations to evaluate the stratification of the Baltic. The chosen stations, resembling stations in Meier (2007), cover the main basins of the Baltic Sea and are displayed in Fig. 10(a). The observational data are in-situ profile data (Copernicus Baltic Sea- In Situ Near Real Time Observations, https://doi.org/10.48670/moi-00032), and the temporal mean is calculated over all time instances between January 1980 and December 2020 where the observational data exist.

Over the whole evaluation period, both simulations tend to fit in-situ observations in upper layers and underestimate temperatures in deeper layers at the monitoring stations Bornholm Deep (SMHIBY5) and Gotland Deep (SMHIBY15), see supplementary Fig. A5. The seasonal cycle is captured by both models. The mean temperature profiles in the Arkona Basin (Arkona, FINO2, Fig. 10b), closely match the observational data for both model results, where a cold bias by ROAM-NBS in this region can be quantified to be $1°C$ at FINO2 and $< 0.5° C$ at Arkona. To a large extent, the standard deviation depicts the interannual variability, which is similar in simulation results and observations.

At station SMHIBY5, which is located in the Bornholm basin, a warm bias in intermediate ocean layers and a cold bias at the sea bottom can be observed, while both model runs coincide well with observational data at the sea surface and differences between coupled and uncoupled simulations are small (Fig. 10b).

Similarly, within the Gotland Deep (SMHIBY15, Fig. 10), both model runs underestimate the mean temperature at depths below 100 m, mostly due to a weak salinity stratification (see Fig. 12) and the overall cold bias at the bottom of the Gotland Deep, will be shown in Sect. 4.1, is confirmed. Here, the coupled model ROAM-NBS again has a larger cold bias than the NEMO-NBS stand-alone run.

The last two stations, SMHIBY31 and SMHISR5C4, lie in the Landsort Deep and Gulf of Bothnia, respectively. At both stations, the ROAM-NBS results depict a warm bias at the sea surface while showing good agreement with observational data below 20 m (Fig. 10). In the SST evaluation,the bias fluctuates around zero over time and is very small in the area of stations SMHIBY31 and SMHISR5C4. Therefore, the limited time resolution of profile data observations can lead to a small positive bias in the mean temperature profiles. The bathymetry does not resolve the Landsort deep, but for the available model depth, the NEMO-NBS stand-alone mean temperature results agree well with the observational data, especially at the sea surface and sea bottom. In the Gulf of Bothnia, the NEMO-NBS results also coincide well with the observational data, depicting a small warm bias in the complete profile: at the sea surface, NEMO-NBS performs better than the ROAM-NBS results, which could be attributed to the minimal better representation of sea ice concentrations in NEMO-NBS within the Gulf of Bothnia, Fig. 9; for lower depths, ROAM-NBS results are closer to observational data.

Overall, the temperature profiles of the NEMO-NBS simulation have a smaller gradient within the mixed layer than the coupled ROAM-NBS results due to the radiation and momentum fluxes exchanged by the atmosphere and the ocean components within the coupled model, leading to a higher coincidence of ROAM-NBS at the mixed layer depth with observational data. At the sea surface and sea bottom, NEMO-NBS stand-alone results show a smaller bias to observational data.





(a)

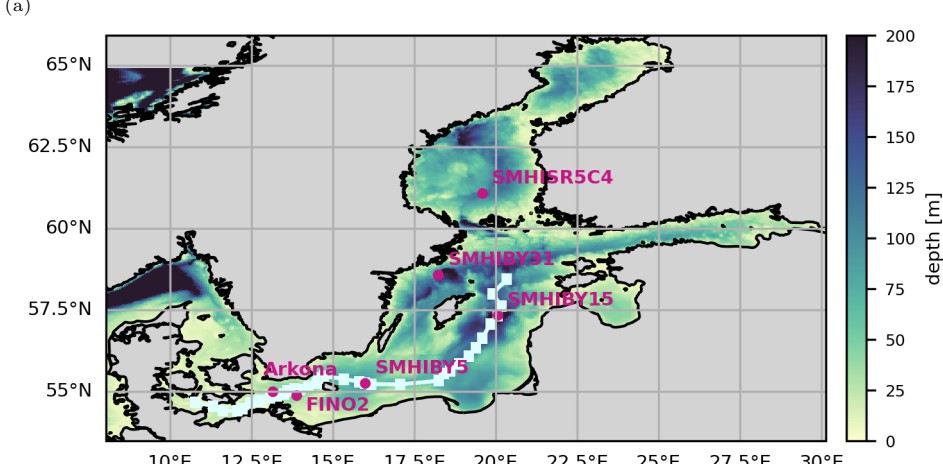

(b)

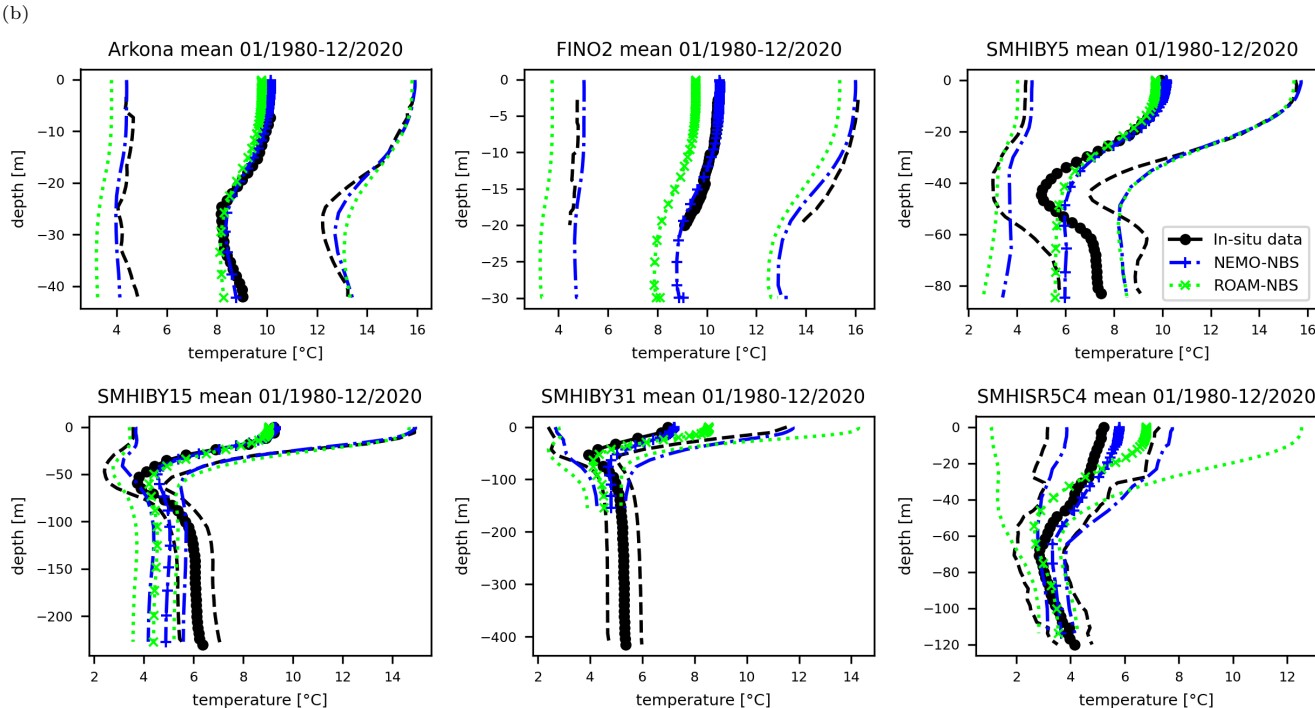

**Figure 10.** Baltic section of applied bathymetry in NEMO-NBS and ROAM-NBS. The stations used for the temperature and salinity mean profile validation (magenta) and the IOW Baltic thalweg measurement path on 11 November 2014 (lightblue) as given in Mohrholz (2016) are displayed (a); mean temperature profiles (January 1980–December 2020) at Baltic stations. Lines with markers: mean data, lines with no marker: mean ± standard deviation (b).





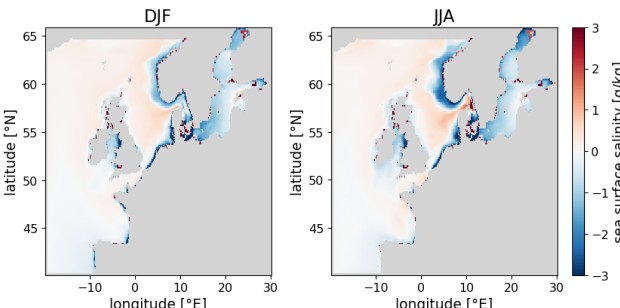

**Figure 11.** Seasonal mean sea surface salinity difference between ROAM-NBS and observations for the period December 1993–November 2020.

### 3.3.3 Salinity

The model sea surface salinity is validated against an interpolated level-4 analysis of the sea surface salinity based on in-situ and satellite observations (Copernicus Multi Observation Global Ocean Sea Surface Salinity and Sea Surface Density, https://doi.org/10.48670/moi-00051). The differences between the simulated mean sea surface salinity of ROAM-NBS and observations for the period December 1993–November 2020 are shown in Fig. 11. These years were chosen as the observational data set is only available for this period. The sea surface salinity tends to be underestimated at the Norwegian and German coasts and the Baltic Sea and tends to be overestimated at the passage from the Baltic Sea to the North Sea. However, both models, ROAM-NBS and NEMO-NBS, show very similar results (Fig. 11). Since the pattern at the coasts seems to be quite constant throughout the year, the cause may originate in a too strong prescribed inflow of fresh water. The underestimation of the salinity in the Baltic Basin may be the result of the general underestimation of the salinity inflow from the North Sea.

The evolution of salinity in deeper ocean layers at the monitoring stations at Bornholm Deep (SMHIBY5) and Gotland Deep (SMHIBY15) is compared in Fig. A6 for both simulations and the Copernicus in-situ profile data. Both models underestimate the salinity in deeper layers but fit observations in higher layers. The representation of the stratification of the Baltic Sea is further analyzed with mean salinity profiles and sections through the Baltic in Sect. 4.1. As an example, one Major Baltic inflow (MBI) event (2014) is chosen for the comparison between observations and simulations. The MBI is reproduced but underestimated at both stations.

As in the Hovmoeller diagrams, the mean salinity profiles between 1980 and 2020 indicate an underestimation of salinity in the deep areas of both simulations (SMHIBY15, SMHIBY5, SMHIBY31, Fig. 12). The surface salinity is well represented at these locations by both ROAM-NBS and NEMO-NBS. At Arkona and FINO2 (Fig. 12), model results highly coincide with the mean observational salinity profiles. The underestimation of bottom salinity could be improved by using longer spin-ups for future simulations. In the Gulf of Bothnia (SMHISR5C4), both simulations show a fresh bias, which is slightly smaller for the NEMO-NBS stand-alone run than for ROAM-NBS. Overall, underestimating bottom salinity leads to a fresher Baltic in the Gulf of Bothnia, as observed at station SMHISR5C4.





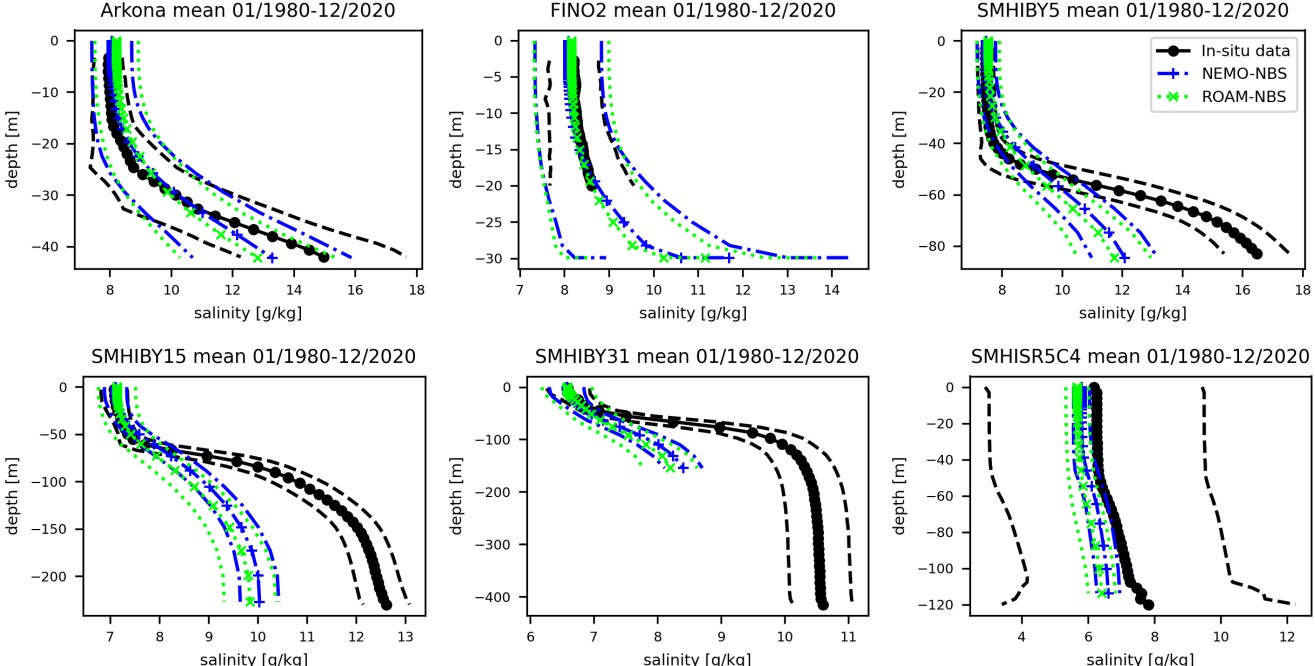

**Figure 12.** Mean salinity profiles (January 1980–December 2020) at Baltic stations. Lines with markers: mean data, lines with no marker: mean ± standard deviation.

### 3.3.4 Sea surface heights

To assess the accuracy of sea surface heights, model results are compared against GESLAv3.0 observational data (Haigh et al.,
2023) within the evaluation period of January 2015 - December 2019. Here, the sea surface height results are bias adjusted by subtracting the mean of SSH within the evaluation period from SSH results. To verify the wind surge component of the water level, the bias adjusted sea surface height is detided with a Demerliac filter. Scatter plots for the detided sea surface height are displayed in Fig. 13 for a station in the German Bight (Cuxhaven), and the Baltic (Landsortnorra). Statistical values such as the number of data points $N$, the correlation coefficient $r$, the root mean square deviation rmsd, and the explained variance $\eta$
are calculated and summarized in Tab. 2 for further stations covering the model domain.

For detided sea surface heights, NEMO-NBS shows a higher correlation at all stations with the observational data than ROAM-NBS, with minimal differences in correlation coefficients. The root mean square deviation of NEMO-NBS is smaller for the majority of the stations. As can be observed in the scatter plots, for detided sea surface height extreme values, especially maxima, ROAM-NBS performs better in the German Bight, where it slightly overestimates maxima at Landsortnorra. The
higher maxima at all stations obtained with the coupled ROAM-NBS simulation indicate a positive effect of the atmosphere-ocean coupling on wind speed quality. NEMO-NBS's performance for sea surface height maxima could be improved by calibrating the wind drag coefficient. Single storm events, which cannot be well represented spatially and seasonally, lead





**Table 2.** Statistical values of detided sea surface height model results vs. GESLAv3.0 observational data evaluated within January 2015–December 2019.

| Station | ROAM-NBS | | | NEMO-NBS | | |
|---|---|---|---|---|---|---|
| | $r$ | rmsd | $\eta$ | $r$ | rmsd | $\eta$ |
| Alteweser | 0.93 | 0.11 | 85 | 0.96 | 0.07 | 93 |
| Bergen | 0.86 | 0.08 | 67 | 0.87 | 0.08 | 69 |
| Borkum | 0.93 | 0.10 | 85 | 0.96 | 0.07 | 92 |
| Bremerhaven | 0.94 | 0.11 | 86 | 0.97 | 0.07 | 95 |
| Cuxhaven | 0.94 | 0.12 | 86 | 0.97 | 0.07 | 94 |
| Eidersperrwerk | 0.94 | 0.13 | 87 | 0.97 | 0.08 | 94 |
| Emden | 0.94 | 0.10 | 86 | 0.96 | 0.07 | 93 |
| Frederikshavn | 0.91 | 0.09 | 74 | 0.92 | 0.07 | 80 |
| Helgeroa | 0.90 | 0.08 | 75 | 0.91 | 0.07 | 80 |
| Helgoland | 0.94 | 0.11 | 85 | 0.97 | 0.07 | 93 |
| Husum | 0.95 | 0.12 | 89 | 0.98 | 0.08 | 96 |
| Landsortnorra | 0.97 | 0.05 | 92 | 0.97 | 0.04 | 94 |
| Leixoes | 0.50 | 0.09 | 25 | 0.48 | 0.10 | 23 |
| List | 0.95 | 0.11 | 88 | 0.97 | 0.07 | 95 |
| Newlyn | 0.67 | 0.11 | 43 | 0.66 | 0.12 | 41 |
| Plymouth | 0.79 | 0.08 | 61 | 0.78 | 0.08 | 60 |
| Rorvik | 0.90 | 0.08 | 79 | 0.91 | 0.08 | 81 |
| Stavanger | 0.85 | 0.08 | 67 | 0.86 | 0.08 | 69 |
| Stpetersburg | 0.95 | 0.09 | 88 | 0.97 | 0.07 | 93 |
| Travemünde | 0.92 | 0.07 | 83 | 0.94 | 0.06 | 89 |
| Tregde | 0.82 | 0.09 | 55 | 0.83 | 0.08 | 61 |
| Vigo | 0.61 | 0.09 | 36 | 0.61 | 0.09 | 36 |
| Viker | 0.91 | 0.08 | 81 | 0.93 | 0.07 | 86 |

to single outliers in the coupled model. These outliers are not present in the reanalysis driven NEMO-NBS results. Overall, the correlation of wind surge of both model results with observational data is high, especially in German national waters. At

stations closer to the domain boundaries, wind surge correlations could be improved. Nevertheless, overall bias-adjusted sea surface height results strongly correlate with observational data (Fig. A7) for the coupled and uncoupled models due to a high tidal component.



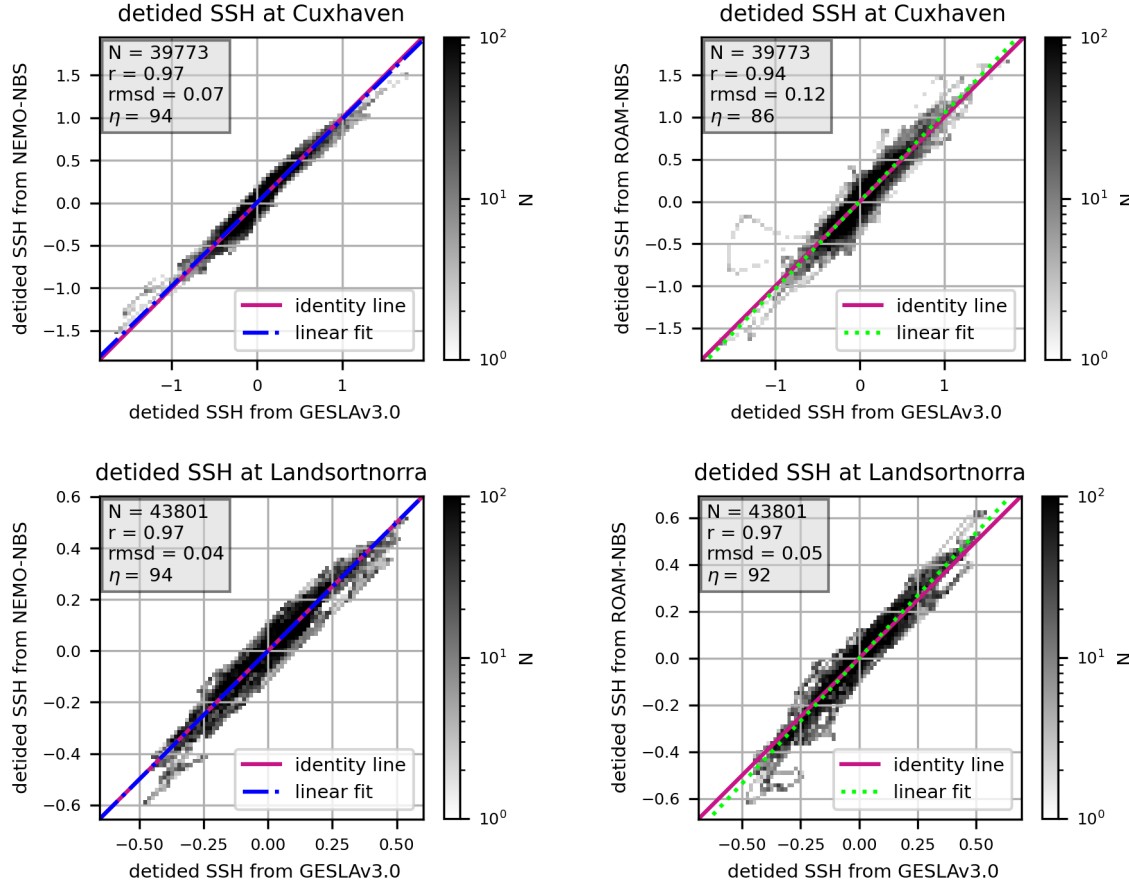

**Figure 13.** Comparison of scatter plots of detided sea surface heights at selected stations in the North Sea and Baltic for NEMO-NBS (blue) vs. ROAM-NBS (green).

## 4 Variability and extreme events

In this section, the model results of ROAM-NBS and NEMO-NBS are evaluated for exemplary extreme events against obser-
vational and reanalysis data. As an adequate representation of the inflow from the North Sea into the Baltic Sea is important
to correctly model the state of the Baltic Sea over longer timescales, a particular focus was put on the analysis of this inflow.
In Sects. 4.1 and 4.2, cross-sections of salinity and temperature through the Baltic and sea surface heights at representative
stations are evaluated around the Major Baltic Inflow (MBI) in December 2014, which was the third-largest recorded inflow.
Its influences are described in Mohrholz et al. (2015). In Sect. 4.3, the model simulation's capacity to track observed marine
heatwaves (MHWs) is examined.





**Figure 14.** Conservative temperature along the Baltic thalweg for NEMO-NBS, ROAM-NBS and observational data Mohrholz (2016) at three time instances around the MBI 2014: 11 November 2014 (top), 7 February 2015 (middle) and 22 March 2015 (bottom).







**Figure 15.** Practical salinity along the Baltic thalweg for NEMO-NBS, ROAM-NBS and observational data Mohrholz (2016) at three time instances around the MBI 2014: 11 November 2014 (top), 7 February 2015 (middle) and 22 March 2015 (bottom).





## 4.1 Stratification along the Baltic Thalweg for the Major Baltic Inflow in 2014

The Baltic Sea is characterized by its strong stratification of temperature as well as salinity, and its correct representation is needed for marine climate applications. To validate the stratification, the Baltic thalweg level 4 dataset (Mohrholz, 2016) is used for temperature and salinity. The interpolated gridded CTD (Conductivity, Temperature, Depth) data set contains the longitudes and latitudes of the stations where measurements were taken, and these slightly vary with each observation cruise. Therefore, the exact coordinates of the bathymetry profiles along the Baltic thalweg in Figs. 14 and 15 may differ from date to date, but the model data is always extracted at the locations given in the observational data. An exemplary observational route is shown in Fig. 10, and the distance along the route to the start is plotted on the x-axis.

To evaluate the model simulation's capability to depict the inflow of saline water into the Baltic, thalweg profiles before the MBI 2014 (November 2014) and after the MBI event (February, March 2015) are displayed for temperature (Fig. 14), and salinity (Fig. 15).

As can be observed in Fig. 14, the temperature thalweg of the coupled ROAM-NBS and ocean stand-alone NEMO-NBS simulation only subtly differs. Before the MBI event, a temperature bias in the deep basins already existed between both model simulations and the observational data. The surface temperature obtained with NEMO-NBS in November 2014 highly coincides with the observational data for $500 \, \text{km} < \text{distance} < 1000 \, \text{km}$, whereas surface temperatures obtained with ROAM-NBS are closer to observational data for $0 \, \text{km} < \text{distance} < 500 \, \text{km}$. Also in the Bornholm basin ($300 \, \text{km} < \text{distance} < 450 \, \text{km}$), ROAM-NBS temperature results are closer to observational data in November 2014. After the MBI, higher temperatures ($>7.5° \, \text{C}$) can be observed in the Bornholm basin and Slupsk Furrow ($450 \, \text{km} < \text{distance} < 550 \, \text{km}$), and partially at the Gotland Basin bottom ($\sim 800 \, \text{km}$) for both simulations. ROAM-NBS and NEMO-NBS both capture this, with NEMO-NBS slightly better representing the bottom signal, though it shows a stronger cold bias in upper layers ($> 500 \, \text{km}$). Overall, the temperature stratification characteristics of the Baltic are depicted well by both NEMO-NBS and ROAM-NBS for the MBI event, although a cold bias in the deep basins is present for both models.

In Fig. 15, the salinity along the thalweg is displayed around the MBI event 2014. In both models, the MBI-related salinity transport is visible but underestimated, particularly in deep basin penetration, cf. Fig. 14. The inflow event can partly be depicted for the Fehmarn Belt, Darss Sill, and Arkona Basin regions. The transport of salty water into the deeper basins is discontinued, resulting in an underestimation of salinity for both NEMO-NBS and ROAM-NBS. Nevertheless, the salinity is already underestimated before the MBI by approximately $2 \, \text{g kg}^{-1}$ in both simulations. The flow from the Slupsk Furrow towards the Gotland Basin is underestimated in both model results. Overall, the NEMO-NBS stand-alone run represents the inflow event better, which can be attributed to differences in the applied surface boundary conditions.

## 4.2 Storm surges in January 2015

During 9–11 January 2015, two consecutive winter storms crossed northern Europe (ELON and FELIX, Haeseler, 2015) and caused much damage and a severe storm surge in the North Sea. Storm ELON passed on 9 January and dissolved afterwards. For 10 January 2015, 12 UTC, the center of storm FELIX with a minimum pressure of less than $960 \, \text{hPa}$ is simulated by







**Figure 16.** Synoptic map for 10 January 2015, 12 UTC, displaying isolines of geopotential at 500 hPa (black lines), mean sea level pressure (white lines) and temperature at 850 hPa (a); wind vectors and wind speed at 10 m (shading) in a smaller area centered around Helgoland (b); time series of wind speed of the DWD weather station on Helgoland and wind speed of ROAM-NBS and ERA5, all at 10 m above ground (c); and (d) time series of maximum sea surface height (SSH) at Helgoland.





ROAM-NBS directly to the west of the Norwegian coast (Fig. 16a). The location and minimum pressure agree well with the

one reported by the DWD surface analysis (Haeseler, 2015). Accordingly, ROAM-NBS simulates strong westerly winds (due to surface friction, near-surface winds are rotated towards the low pressure center compared to the geostrophic wind) on 10 January with a maximum of up to $24\,\mathrm{m\,s^{-1}}$ (about $86\,\mathrm{km\,h^{-1}}$) to the northwest of Helgoland (Fig. 16b). Observed winds on Helgoland show maxima of up to $20\,\mathrm{m\,s^{-1}}$ on 9 and 10 January and slightly weaker maxima on 8 and 11 January (Fig. 16c), indicating that storm conditions prevailed over several days. The near-surface wind speed is not as well matched with the

observations by ROAM-NBS as by ERA5 (Fig. 16c), but the maximum on 10 January is well reproduced by both.

Associated time series of the sea level minus the yearly mean sea level during storm events ELON and FELIX are presented in Figs. 16d and A8. The storm events in January occur shortly after the major Baltic inflow event in December 2014. The non-detided SSH results are compared against GESLAv3.0 observational data described in Haigh et al. (2023). Stations in the German Bight (Helgoland), Skagerrak (Helgeroa), and Baltic Sea (Travemünde) are chosen to discuss the model's instant

behavior exemplarily. At the station Helgoland, the warning level of 2.5 m for a severe storm surge in the North Sea is exceeded on 11 January in the early morning hours. A small time shift can be observed compared to the wind speed maximum on 10 January. The maximum sea level on 11 January is slightly better represented by the coupled model than NEMO-NBS, although wind speed maxima are comparable in ROAM-NBS and ERA5. Further, the coupled model better coincides with the displayed maxima, which fits the results obtained in the scatter plot at Cuxhaven Fig. 13. The better representation of the SSH maxima

in ROAM-NBS can mainly be attributed to the differences in the treatment of surface boundary conditions, especially the calculation of the wind stress by the surface momentum fluxes, cf. Sect. 2.3.

In the Skagerrak, again, the maximum sea level for storm events ELON and FELIX in January 2015 is better represented by the coupled model, where some of the minimums in the time series are better represented by NEMO-NBS.

At station Travemünde, no high sea level maximum was present in January 2015. Higher amplitudes around the maximum sea

level (4 January) as well as lower sea level values around the sea level minimum (3 January) can be observed for ROAM-NBS in comparison to NEMO-NBS.

### 4.3 Marine heatwaves

Marine heatwaves (MHWs) are discrete periods of anomalously high sea surface temperatures. Following the widely used definition by Hobday et al. (2016), MHWs are identified as periods of at least five consecutive days during which temperatures

exceed the 90th percentile of a baseline climatology. To detect MHWs in data and model output, we apply the open-source Python package developed by Oliver (2016) to sea surface temperature data extracted at the location of the long-term monitoring station "Lighthouse Kiel" in the western Baltic Sea (10.267°E, 54.5°N).

To evaluate model performance in terms of reproducing extreme events, we compare three standard MHW metrics—the annual number of events, the maximum intensity, and the total number of MHW days per year—across four datasets: the two model

configurations, in-situ observations, and reanalysis data (Copernicus Baltic Sea Physics Reanalysis, https://doi.org/10.48670/moi-00013; Atlantic- European North West Shelf- Ocean Physics Reanalysis, https://doi.org/10.48670/moi-00059) at two locations. The locations are chosen based on the availability of long-term (> 30 years) observational data. One station is in the western





**Figure 17.** Annual marine heatwaves metrics computed for the location Lighthouse Kiel in the western Baltic Sea from in-situ data at 0.5 m depth (black), Copernicus Baltic Sea Physics reanalysis data (orange), NEMO NBS simulation (blue) and ROAM NBS simulation (green). Common climatology period is 1993 to 2021. The metrics compared here are number of MHW events (a), maximum intensity [°C] (b) and total days of MHW conditions (c). The green stars indicate the years where reanalysis data is not available, which starts in 1993. The black stars indicate years with too large data gaps in the in-situ data (1999, 2002, 2015, 2016).





Baltic Sea (Station Lighthouse Kiel, 10.27 lon, 54.4 lat) and one in the German Bight (Station UFS German Bight, 7.45 lon, 54.17 lat). All MHW metrics are computed relative to each dataset's own climatology, using the common baseline period from

1993 to 2021. Supplementary Fig. A10 compares the seasonal climatology and corresponding 90th percentile threshold across datasets for the station LT Kiel. While the model shows a cold bias in winter and a warm bias during summer, the MHW detection is not affected by this, as it is performed relative to each dataset's individual climatology. Figure 17 presents a comparative analysis of annual MHW metrics at Lighthouse Kiel derived from in-situ observations (black), reanalysis data (orange), and the two NEMO model configurations: NEMO-NBS (blue) and ROAM-NBS (green), spanning the period 1989–2021. Panel

(a) shows the annual number of MHW events, panel (b) illustrates the maximum intensity of MHWs (in °C) in each year, and panel (c) displays the total number of MHW days per year. The same MHW evaluation is presented in Figure A9 for the location UFS German Bight.

Overall, all model configurations capture the inter-annual variability in MHW characteristics reasonably well at both locations. The NEMO-NBS (blue) generally aligns a little more closely with the observational data in terms of both event frequency, max-

imum intensity, and duration, particularly in recent years. At both stations, the ROAM-NBS simulation has a slightly higher correlation with the observed events and days. However, discrepancies are observed in certain years where model simulations either overestimate or underestimate the magnitude and extent of MHWs. For example, at Lighthouse Kiel, in 2014 and 2018, both NEMO-NBS and ROAM-NBS tend to overestimate MHW metrics relative to observations, particularly in terms of total days and maximum intensity.

Despite some variability, the model simulations demonstrate skill in reproducing the temporal patterns and intensities of MHWs observed in the region, supporting their application for understanding past and projecting future marine heatwave conditions.

## 5 Conclusions

Evaluation results from the ERA5/ORAS5-driven evaluation simulation of the coupled regional ocean–atmosphere model ROAM-NBS were presented. ROAM-NBS will be used to produce regional climate projections, which will contribute to the

EURO-CORDEX ensemble. Therefore, ROAM-NBS and the individual stand-alone versions of the ocean (NEMO-NBS) and the atmosphere (ICON-CLM) were assessed with respect to different observations and to reanalyses. NEMO-NBS as well as ROAM-NBS exhibit a small SST bias, which is on area-average below ±0.5 K. For individual seasons and regions, it can also reach larger values. Especially over the Atlantic Ocean, a cold bias prevails for all seasons except summer. The SST bias is only slightly increased in summer in ROAM-NBS compared to NEMO-NBS by about 0.25 K. For both ROAM-NBS and NEMO-

NBS, there is no increase of the bias with time throughout the evaluated period of 1979–2020. Warming trends in the North and Baltic Sea are well reproduced by the simulations. The surface temperature difference against ERA5 exhibits clearly larger values over land than over the ocean. The near-surface air temperature bias over land is overall negative for both ICON-CLM and ROAM-NBS, with a small overestimation of the diurnal minimum temperature and a more pronounced underestimation of the diurnal maximum temperature in all seasons. The temporal evolution of mean temperatures over land is generally in

agreement with observational data and reanalyses, with the largest cold bias in Spain and Portugal especially after 1995 and a



warm bias in the drier, more continental region of south-east Europe. Differences between ROAM-NBS and ICON-CLM are very small in all land regions and years.

Over the ocean, the SST differences between ROAM-NBS and ICON-CLM reflect the bias of NEMO-NBS. The sign of the SST difference coincides with the sign of the differences for sensible and latent heat flux, wind speed, and precipitation, i.e. in regions and seasons where the SST is higher in ROAM-NBS than in ICON-CLM, also the heat fluxes, wind speed, and precipitation are higher and vice versa. This relationship means that the SST bias introduced into the coupled system by NEMO-NBS directly influences the atmospheric fields over water, but as shown before, this does not have a systematic influence on the land areas. Compared to station observations, wind speed over the ocean is underestimated, which is slightly more pronounced in ROAM-NBS than in ICON-CLM due to the SST cold bias along the German coast in ROAM-NBS, which causes a reduction of wind speed compared to ICON-CLM. Future improvements in NEMO-NBS could include a time-dependent chlorophyll field that leads to a season-dependent absorption of radiation by the ocean. Improved radiative forcing at the ocean surface could reduce the SST bias in all seasons. Further, a calibration of the lateral and vertical diffusion parameters could enhance the transport over steep ridges in the bathymetry and therefore weaken the salinity bias. In the coupled system, it could be an option to send the ocean albedo over water to the atmospheric part, but then an adaptation in the NEMO coupling interface would be necessary.

The comparison of seasonal mean sea ice concentration between the NEMO-NBS and ROAM-NBS simulations and observational datasets reveals that both simulations tend to overestimate sea ice concentration, particularly during the spring season. While the simulations show good agreement with observations in winter, they significantly amplify sea ice extent in the Gulf of Bothnia during spring. This discrepancy is likely linked to an underestimated salinity in the region. Additionally, the lack of ice dynamics in the current model configurations may contribute to these inaccuracies. Incorporating dynamic ice processes alongside thermodynamic ones and parameter tuning of the thermodynamic ice model may improve the models' performance and alignment with observed sea ice behavior.

The validation of modeled sea surface salinity (SSS) against observational data from December 1993 to November 2020 reveals consistent spatial patterns and systematic biases in both ROAM-NBS and NEMO-NBS simulations. While surface salinity is generally well captured in the open Baltic Sea and certain coastal regions, persistent underestimations are observed near the Norwegian and German coasts, as well as within the Baltic Sea. Conversely, SSS is overestimated at the transition zone between the Baltic Sea and the North Sea. These biases may be attributed to overly strong prescribed freshwater runoff and insufficient representation of saline inflow from the North Sea. In deeper layers, both models consistently underestimate salinity, particularly in the Baltic basins, although the surface layers show good agreement with observations. The stratification structure and major inflow events are qualitatively reproduced but quantitatively underestimated. Overall, while the models effectively capture large-scale salinity patterns and seasonal behavior, further refinement of boundary conditions and freshwater forcing is necessary to improve deep water salinity representation and coastal accuracy.

The comparison of mean temperature and salinity profiles from the ROAM-NBS and NEMO-NBS simulations against in-situ observational data reveals both strengths and limitations in the models' ability to reproduce stratification in the Baltic Sea. While both simulations capture surface conditions well across most stations, discrepancies become more apparent at depth.





ROAM-NBS exhibits a notable cold bias in bottom waters, especially in deeper basins like Gotland Deep. Both simulations underestimate bottom salinity in the central and northern Baltic, contributing to an overall fresher modeled Baltic Sea. For future simulations with NEMO-NBS or ROAM-NBS, longer spin-ups of at least 10 years of the ocean component shall be applied to enhance simulation results. Surface salinity, however, is generally well represented. The ROAM-NBS coupled simulation improves performance in capturing the mixed layer temperature gradient, though NEMO-NBS often achieves lower overall temperature and salinity biases. These findings suggest that while coupled atmosphere-ocean modeling, as in ROAM-NBS, offers advantages for capturing specific vertical structures, future work should focus on improving the ocean component's ability to model inflow events into the deep basins.

The comparison of model simulations with GESLAv3.0 observational data for the period of January 2015 to December 2019 shows a generally strong agreement in sea surface height (SSH) and wind surge accuracy. After bias adjustment and detiding, the NEMO-NBS simulation shows a slightly better correlation with observations across most evaluated stations than the ROAM-NBS simulation. However, ROAM-NBS captures extreme values more effectively, particularly maxima in the German Bight and Kattegat/Skagerrak regions, suggesting enhanced wind forcing through atmosphere-ocean coupling.

By comparing key marine heat wave (MHW) metrics across observations, reanalysis, NEMO-NBS, and ROAM-NBS, we find that both simulations capture inter-annual variability well, with NEMO-NBS showing better alignment with observations in recent years. Although some overestimations occur in specific years, particularly for intensity and duration, the overall strong correlations support the models' ability to simulate MHW dynamics.

Overall, the coupled simulation ROAM-NBS provides satisfactory results for both the ocean and the atmosphere when compared to observations and reanalysis-driven stand-alone simulations. Therefore, it can be applied to compute regional climate projections for Europe and further deliver climate adaptation information for German national waters. Since ROAM-NBS does not depend on surface boundary conditions from regional climate projections, unlike NEMO-NBS, CMIP6 information can be downscaled more efficiently when using the coupled model. Moreover, the regional coupled climate projections will be beneficial also for the atmosphere as the ocean part of ROAM-NBS delivers higher resolved and probably more accurate information at the surface of the North and Baltic Sea than CMIP6 GCM climate projections.

*Code and data availability.* For ICON, the open-source release 2024.07 (https://gitlab.dkrz.de/icon/icon-model/-/tree/release-2024.07-public, last access: 15 July 2025) was used. The model version including OASIS interfaces can be accessed at https://github.com/vmaurerDWD/icon-ROAM (last access: 15 July 2025). For NEMO, the open-source release 4.2.0 was used (https://forge.nemo-ocean.eu/nemo/nemo/-/tree/4.2.0, last access: 15 July 2025). All source codes (ICON, NEMO, OASIS3-MCT, and XIOS) are additionally stored on Zenodo (https://doi.org/10.5281/zenodo.17035585; Meyer et al., 2025a). Run scripts and configurations necessary to perform the ICON-CLM, NEMO-NBS, and the coupled simulation are also available on Zenodo (https://doi.org/10.5281/zenodo.17037353; Meyer et al., 2025b). Necessary input data for a short time period and additional post-processing scripts can be found under the same resource.

DWD station data are available via CDC (https://cdc.dwd.de/portal/](https://cdc.dwd.de/portal/). Measurement data of FINO1 are freely available after registration via https://www.bsh.de/DE/DATEN/Klima-und-Meer/Meeresumweltmessnetz/_Module/Info_Stationen/info_fino1_



node.html (last access: 15 July 2025). E-OBS data v27.0 were obtained from https://surfobs.climate.copernicus.eu/dataaccess/access_eobs.

php#datafiles (last access: 15 July 2025).





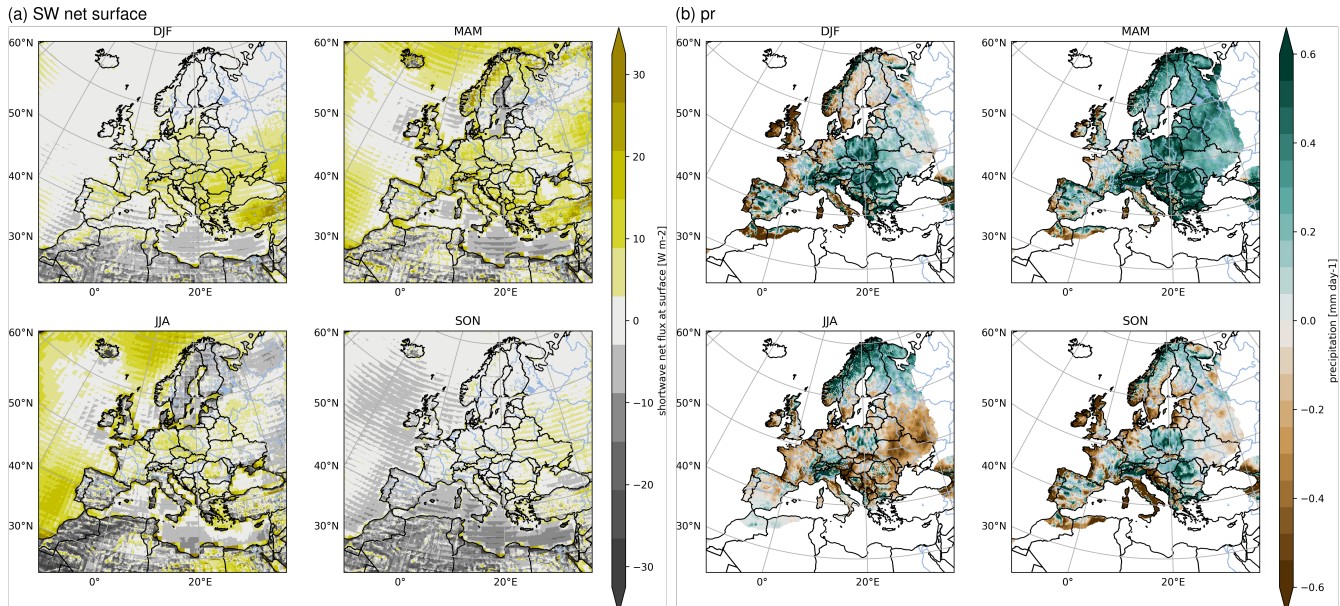

**Figure A1.** Seasonal mean biases of surface shortwave net radiation for ROAM-NBS against CERES, averaged over 2001–2020 (a) and of precipitation against E-OBS, averaged over 1979–2020 (b).

## Appendix A: Complementing evaluations

### A1 Mean meteorological conditions

As mentioned in 3.1, a cloud cover tuning is applied in ICON-CLM to reduce an overestimation of downward shortwave radiation at the surface (Geyer et al., 2025). The seasonal mean surface shortwave net radiation bias against a satellite product
(CERES) is shown in Fig. A1a. After the tuning, it is not much larger than $\pm 10\,\mathrm{W\,m^{-1}}$. Largest values can be observed in summer (JJA). As the tuning was not location-dependent, it was not possible to decrease the bias further.

To give an estimate for the absolute precipitation bias of ROAM-NBS and ICON-CLM, respectively, the mean seasonal biases against E-OBS are given in A1b. In comparison, the differences between the two simulations (6a) are very small.

Complementing Fig. 5, the seasonal bias maps for ROAM-NBS against E-OBS are shown in Fig. A2 for tasmin and tasmax.
They confirm that the tasmin bias is small and non-systematic compared to the more systematic cold bias for tasmax. One slightly outstanding pattern in the tasmin bias is red pattern in Scandinavia in winter (DJF). However, it is unclear if this also might be over-emphasized by the comparison against E-OBS as it does not show up that clearly when comparing against ERA5 (not shown).

To give an insight into the temporal evolution of temperatures within the evaluation period, time series of yearly averaged tas
are given in Fig. A3 for different countries. In some cases, neighbouring countries were combined into one time series (GB and Ireland, Spain and Portugal, Norway and Sweden, and the Baltic states including Estonia, Latvia and Lithuania). In general,





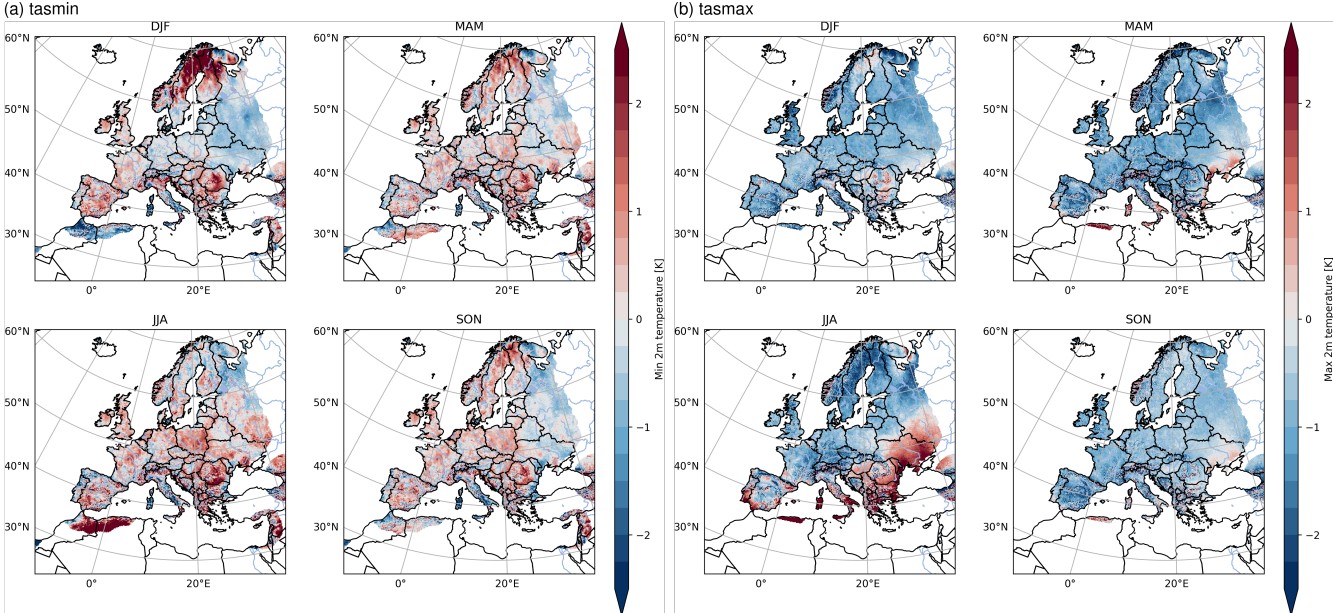

**Figure A2.** Seasonal mean biases of daily minimum and maximum temperature of ROAM-NBS against E-OBS, averaged over 1979-2020.

both ICON-CLM and ROAM-NBS reproduce the mean trends as well as the year-to-year variability of the references (E-OBS and ERA5) very well. As shown before, a cold bias can be discerned which is largest in Spain and Portugal and also clearly visible for GB and Ireland. In the eastern part of the domain, e.g. in Ukraine, the summertime warm bias is dominant. In all cases, ROAM-NBS and ICON-CLM are very similar.

### A2 Mean ocean conditions

To complement the evaluation of SST bias evolution in winter and summer, biases and absolute time series for SST integrals over the same regions (whole domain, Open Atlantic, North Sea, Baltic Sea) are presented for all seasons in Fig. A4. In support of the mean profile validation of salinity and temperature, Hovmöller diagrams are provided in Figs. A5 and A6, to illustrate the temporal evolution of these variables at stations SMHIBY5 and SMHIBY15. Furthermore, extending the analysis of detided sea surface height (SSH), Fig. A7 presents a scatter plot comparison of bias-adjusted SSH at the tidal station Plymouth for NEMO-NBS versus ROAM-NBS. Both model configurations exhibit very high correlation with the GESLAv3.0 observational dataset.

### A3 Variability and extremes

Time series within the same time period as reviewed in Sect. 4.2 are provided for two additional stations: Helgeroa and Travemünde. The results show a good agreement of ROAM-NBS and NEMO-NBS in the Skagerrak and Baltic Sea, where the maximum SSH is better represented by ROAM-NBS at Helgeroa, whereas at Travemünde the SSH maximum is overestimated





**Figure A3.** Yearly time series of tas for different countries / regions for ROAM-NBS, ICON-CLM, E-OBS and ERA5.





(a) absolute SST, Baltic Sea

(b) SST bias, Baltic Sea

(c) absolute SST, North Sea

(d) SST bias, North Sea

(e) absolute SST, Open Atlantic

(f) SST bias, Open Atlantic

**Figure A4.** Seasonal mean area-mean SST time series for NEMO-NBS, ROAM-NBS, and Copernicus observation data for the years 1981–2020 for the Baltic Sea (a), the North Sea (c), and the Open Atlantic (e). Difference of seasonal SST time series for NEMO-NBS and ROAM-NBS, respectively, to Copernicus observation data for the Baltic Sea (b), the North Sea (d), and the Open Atlantic (f).



(a) In-situ data      (b) ROAM-NBS      (c) NEMO-NBS

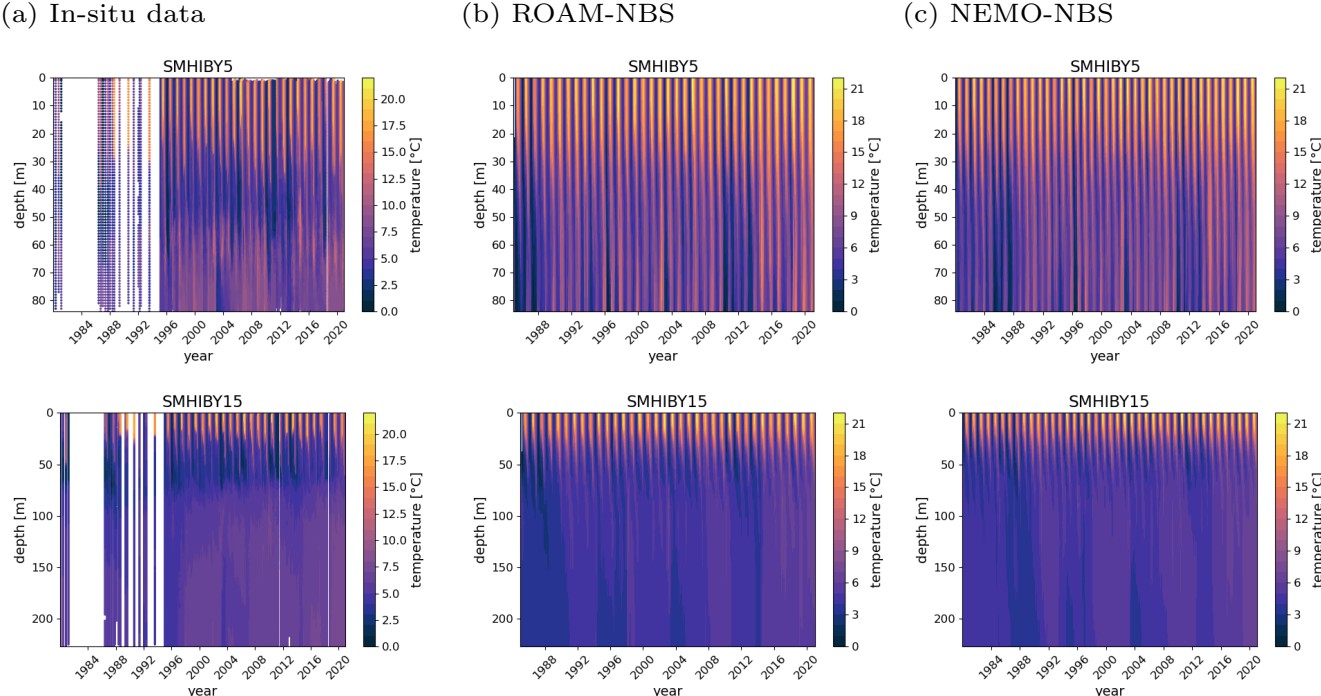

**Figure A5.** Observed (left) and simulated temperature (ROAM-NBS (mid), NEMO-NBS (right)) as a function of depth and time at the monitoring stations Bornholm Deep (SMHIBY5) and Gotland Deep (SMHIBY15).

by both NEMO-NBS and ROAM-NBS with a slightly better fit by NEMO-NBS. The marine heatwaves are additionally evaluated in Fig. A9 at the station UFS German Bight to assess NEMO-NBS's and ROAM-NBS's performance in the North Sea. In

Fig. A10, the seasonal climatology and corresponding 90th percentile threshold are presented for ROAM-NBS, NEMO-NBS, the Copernicus Reanalysis and in-situ observation data for the station Lighthouse Kiel.





(a) In-situ data    (b) ROAM-NBS    (c) NEMO-NBS

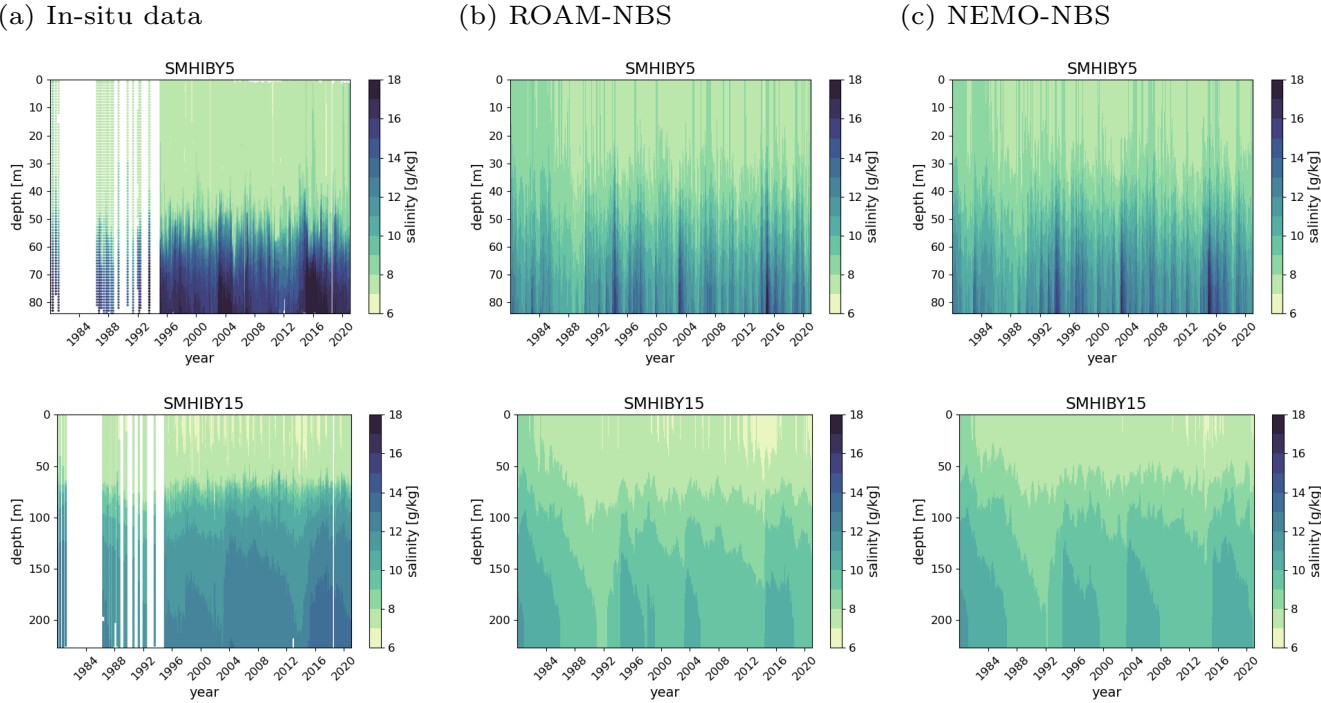

**Figure A6.** Observed (a) and simulated salinity (ROAM-NBS (b), NEMO-NBS (c)) as a function of depth and time at the monitoring stations Bornholm Deep (SMHIBY5) and Gotland Deep (SMHIBY15).

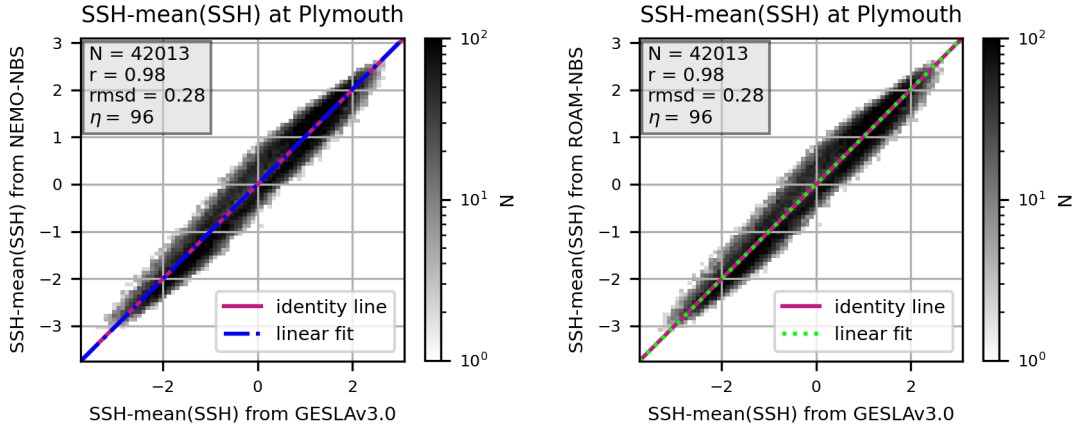

**Figure A7.** Comparison of exemplary scatter plots of bias adjusted sea surface height at a station in a tidal area (Plymouth) for NEMO-NBS (blue) and ROAM-NBS (green).





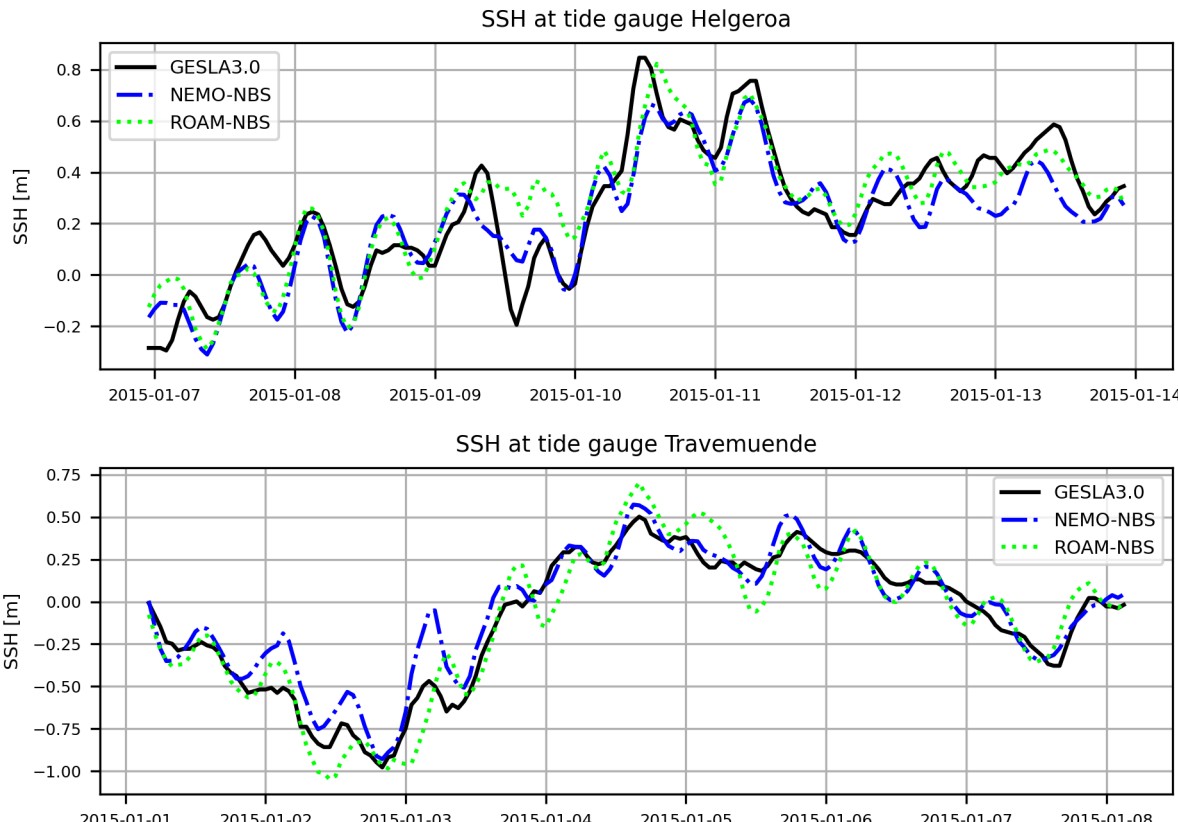

**Figure A8.** Comparison of model results and observations for maximum sea surface height in January 2015 at selected stations in the Kattegat and Baltic.



**Figure A9.** Annual marine heatwaves metrics computed for the location UFS German Bight in the North Sea from in-situ data at 0.5 m depth (black), Copernicus Atlantic- European North West Shelf- Ocean Physics reanalysis data (orange), NEMO NBS simulation (blue) and ROAM NBS simulation (green). Common climatology period is 1993 to 2021. The metrics compared here are the number of MHW events (a), maximum intensity [°C] (b) and total days of MHW conditions (c). The orange stars indicate the years where reanalysis data is not available, which starts in 1993. The black stars indicate years with too large data gaps in the in-situ data (1990, 2007, 2010, 2014, 2016, 2018, 2019, 2020).



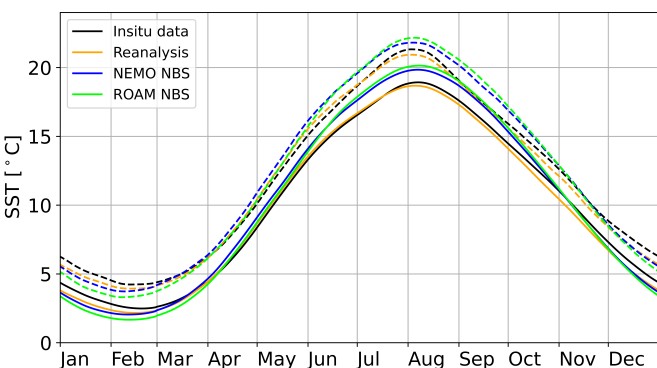

**Figure A10.** Seasonal climatology (continuous) and corresponding 90th percentile threshold (dashed) computed for the location Lighthouse Kiel in the western Baltic Sea from in-situ data at 0.5 m depth (black). NEMO-NBS simulation (blue) and ROAM-NBS simulation (green) and Copernicus Reanalysis (orange). Common climatological period is 1993 to 2021.



*Author contributions.* VM set up and performed the coupled and the atmosphere-only simulations, with contributions by HTMHH. JM, WDW, FJ, and BME set up the ocean simulation with contributions by JS, and JM and WDW performed the simulation. VM, WDW, RB, JM, and CH analyzed the simulations and wrote the article together with the contributions by all authors.

*Competing interests.* The authors declare that no competing interests are present.

*Acknowledgements.* We acknowledge the contributions by E. Maisonnave (CERFACS) within the project IS-ENES funded by the European Union's Horizon 2020 research and innovation programme under grant agreement No 824084.

We acknowledge Carsten Viergutz and Claudius Fleischer from the Bundesanstalt für Gewässerkunde (BfG) for providing the runoff dataset, and Lars Axell from the Swedish Meteorological and Hydrological Institute (SMHI) for supplying hindcast data used in the ocean model

initialization.

This work used resources of the Deutsches Klimarechenzentrum (DKRZ) granted by its Scientific Steering Committee (WLA) under project ID bb1338. Helmholtz-Zentrum Hereon is coordinating the Coastal Futures project (https://www.coastalfutures.de, last access: 3 July 2025) within "Deutsche Allianz für Meeresforschung", which was founded by Germany's federal government and the northern German states. It brings together the leading institutions in German marine research. DWD and BSH are associated partners of Coastal Futures. ChatGPT and

Perplexity were used to support the writing of evaluation scripts. We acknowledge the E-OBS dataset from the EU-FP6 project UERRA (http://www.uerra.eu) and the data providers in the ECA&D project (https://www.ecad.eu).



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
