# Peer review of "Evaluation of coupled and uncoupled ocean-ice-atmosphere simulations using icon\_2024.07 and NEMOv4.2.0 for the EURO-CORDEX domain"

_EGUsphere, 2025_

## Referee Comment (RC1)

Review of the manuscript egusphere-2025-3407:
Evaluation of coupled and uncoupled ocean-ice-atmosphere simulations using icon_2024.07 and NEMOv4.2.0 for the EURO-CORDEX domain
by Vera Maurer, Wibke Düsterhöft-Wriggers, Rebekka Beddig, Janna Meyer, Claudia Hinrichs, Ha Thi Minh Ho-Hagemann, Joanna Staneva, Birte-Marie Ehlers, and Frank Janssen

The manuscript describes the evaluation of a new atmosphere-ocean coupled model and the two respective atmospheric and oceanic stand-alone models on the ERA5/ORAS5 period, on the EURO-CORDEX domain. The setup of the coupled model is well described. The results are often compared to observations, showing that they are reasonably realistic, and that the coupled model can later be used to perform historic and scenario simulations. The evaluation is more detailed on the oceanic part, which is very interesting per se, but I think it could be a bit enlarged on the atmospheric part.
I find the manuscript well written. The most serious restriction I have about the study concerns the period, which is often inconsistent with the 1979-2021 one mentioned on the first line of the abstract, and the 1979-2020 one on line 216, without any explanation. Indeed the other periods are:
- (1) 1979-2020 on the overview of the experiment (Line 216)
- (2) 1983-2020 for SST from ROAM-NBS except for marine heat waves (MHW from 1989 on)
- (3) 1981-2020 for SST from NEMO-NBS except for MHW (I understand 1981 is chosen for the comparison to Copernicus data, but the 1rst years could be shown at least on fig. A4)
- (4) 1989-2021 for MHW (1989 coinciding with observations)
- (5) 1979-2020 for atmospheric variables
- (6) 1979-2020 in fig. 7 with also the skin temperature
If (2) is an effect of ocean spin-up (and even 1984 seems out of range in fig. 3 and A4), then the atmospheric variables should not be shown before.
(4): There is no reason to add 2021 for the MHW only, without showing this year for the other diagnostics.
So to my opinion, one option is to explain why the first years of ROAM-NBS and NEMO-NBS are not shown, and then only show the same period for the atmospheric ROAM-NBS; the second option is to announce a 1983-2020 evaluation simulation.
With this minor revision, I believe the article will be ready for publication.

Please find below some more specific remarks, suggestions and corrections:
- abstract: it should be precised that it is an ERA5/ORAS5 simulation;
- L12: in the abstract and other parts in the manuscript, the authors say that in the coupled model, the SST bias leads to biases in the ocean-atmosphere fluxes: I don't agree with that. A coupled model develops it's own state of equilibrium, SST and atmosphere-ocean surface are linked.
- L28: the delay behind global simulations is also true for regional downscaling of the atmosphere, because they are also forced by global simulations;
- L57: in the manuscript one can find either SI3 or SI$^3$: it should be normalized;
- The new CORDEX Task Force on Regional Climate Projections ([https://cordex.org/strategic-activities/taskforces/task-force-on-regional-ocean-climate-projections/](https://cordex.org/strategic-activities/taskforces/task-force-on-regional-ocean-climate-projections/)) could be mentioned in the Introduction.
- L98: add Northern to "its adjacent seas" (there is no Mediterranean or Black Sea);
- L137: I think that the tuning mentioned in L262-263 should be presented here;
- L138: "For ROAM-NBS as well as for UDAG" : add "and ICON-CLM"
- L138: change icon-2024.07 for icon_2024.07 (as in the title and L56)
- L145: two ((
- L147: the lateral resolution is 2nm?

- L148: could the authors precise more the vertical distribution of the 50 layers? Which is the depth of the first ones?
- L149: 2 "chosen"
- L151: the reference is Madec et al. and not Gurvan et al. (to be corrected in the bibliography as well);
- L182: Craig et al. is the reference for OASIS3-MCT_3.0: the manual of OASIS3-MCT_5.0 by Valcke et al. 2021 could be added;
- L224: is it the same ocean restart for ROAM-NBS?
- L245 and followings: replace sea surface temperature by SST;
- L247: precise the Copernicus data period;
- L253: the authors chose the same seasons as in atmospheric studies, but for information the seasons usually used for ocean variables are JFM, AMJ, JAS, OND;
- L263: cf. L137 above
- L267-268: this parametrisation should be discussed in §2.2 or 2.4;
- L289: don't put "surface air temperature": it is confusing with surface temperature above;
- Figure 6: the "S" of seasonal is missing;
- §3.2.2: I think LongWave fluxes should be shown as well; and compare to the observations, not only RAOM-NBS to ICOM-CLM, which can be biased as well;
- Fig.7 and in the text: I don't understand how the authors choose now the skin temperature: do they have it as an output of NEMO4 (and from 1979?)? The SST comparison to ERA5 SST (which is the one imposed to ICON-CLM) is enough, as it was compared to Copernicus before.
- L316: I don't like the formulation "the skt difference determines the sign of the flux…"; as I said before they are linked in the coupled model; besides they qualify their statements on L325;
- L389: "which" will be shown;
- L393: a space is missing between "evaluation," and "the bias";
- L413-414: indeed the sea surface salinity is highly linked to the E-P-R flux; concerning the runoff, there are options in NEMO which for example propagate the runoff through the vertical, and also to enhance the vertical mixing at the river mouths: the choice here could be precised in § 2.2 or 2.4, or it could be discussed here;
- L425: the restart of the simulation can sometimes also explain some biases in the deeper layers;
- L433: add fig.8 for Cuxhaven;
- L435: add "nearly" at "a higher correlation at all station" because it's not the case for all;
- L444: I'm not a specialist, but is "wind surge" appropriate here when the authors show only the SSH?
- Fig. 13: replace "blue" by "left" and "green" by "right";
- Fig. 14: what are the isolines for?
- L465: it would be of interest to explain what is the "normal" situation of the inflow in the Baltic;
- Fig. 16 c: add ICON-CLM?
- L517: I guess Lighthouse Kiel is also Leuchtturm Kiel of fig.8: choose the same name;
- Fig. 17: Orange stars, not green; it would be interesting to show the MHW from the beginning of the simulations, even if there are no observations;
- L523: idem L517, and add Cuxhaven of fig. 8 for UFS German Bight (if I'm right?)
- L536: the authors must compute the correlations to say that;
- Chapter 4: at the end of this chapter it would be nice to add a conclusion;
- L582: the imposed runoff comes from observations, so do the authors think they might be too strong? cf. Remark for L413-414, and also there could be a discussion about coupling the runoff with a hydrological model: it is the best solution for future scenario simulations. Besides the E-P budget is also of much importance in the surface salinity;
- L593: indeed the spin-up period is very important for the ocean, but also for the coupled model to reach its equilibrium, and as I said before concerning the period of the study, the authors must explain if they think that the 1rst years of the ROAM-NBS simulation is in a spin-up phase.

- L607: the authors don't show the computed correlations;
- L633: add Fig. Before 6a
- Fig. A5: replace "left, mid, right" by "a, b, c"
- Fig. A7: replace "blue" by "left" and "green" by "right"

---

## Referee Comment (RC2)

The preprint represents an evaluation of ROAM-NBS (ICON + NEMO v4.2 + SI3, coupled via OASIS3-MCT, alongside the uncoupled components for the period 1979 - 2020. The mean climate in the atmosphere in generally well reproduced with remaining issues concentrating on an observed SST bias, sea-ice overestimation in the Baltic Sea and too low salinities in the deeper basins in the Baltic Sea impacting stratification and inflow representation. The study itself is timely and highly relevant. The paper is well written. Find suggested revisions below:

**Major concerns:**

In the beginning of the paper you stress that the availability of regional ocean projections is sparse and emphasize that the focus of the evaluation will be on the Baltic Sea region.

However, I find that the representation of the Baltic Sea dynamics is not yet satisfactory. In my view, there are two possible ways forward:

- 1. Recalibrate the model which I know is a long and stressful approach.
- 2. If the model cannot yet capture key dynamics (beyond surface variables) in the Baltic Sea, I suggest explicitly stating that the system is not yet production-ready in that regard and that further improvement is needed.

Alternatively, if I am mistaken, I welcome further clarification. My intent is not to criticize the effort but to ensure the results are accurately contextualized. This is in no way an offense to the great work that is being presented here.

I will try to explain my concerns below along with the text.

Line references:

L148: Please clarify the implementation of the  $\sigma$ – $z^*$  grid. At which depth or criterion switches the model between  $\sigma$  and z coordinates.

L191: 2x respective

L197: Please be consistent with SI3 or SI3

L199: How is the albedo over the water set?

Table 1: Why is rain bold?

L218: Maybe some overview figure/table addressing the different evaluation periods would be good. I lose track throughout the article.

L222: Is 4 years of spin up really enough for the Baltic Sea. I would be really interested to see the timeseries of stations BY2, BY5, BY15 for surface salinity and bottom salinity (similar to Hordoir et al., 2019 their Figure 8).

L234: The sentence is not so easy to understand. It sounds strange to calibrate fixed reanalysis data. I guess you mean to calibrate the initial/boundary conditions extracted from the reanalysis?

Figure 2 (a) if the bias of the SST is locally up to 1.5K, please adjust the colorbar. Please also consider using discrete steps. In addition, the coloring of the contour lines are hard to follow.

L251: I am not exactly sure why the bias of the SST in the northern part cannot be discussed. If you mean that you cannot compare it because ROAM-NBS is overestimating the sea ice, fixing the SST whereas the OBS show different temperatures, please just mask this area.

Figure 3: I am not sure if an average is suitable here, because you mix positive and negative anomalies. I feel that something like a RMS would be better here, but I am not convinced myself.

L263: What kind of tuning is applied. You mean expert tuning to adjust certain cloud parametrization schemes? Please name it shortly.

L271: What is the reason for the decrease? If not discussed, you can also delete it.

L274: Many repetitions of time series

Figure 6: Typo at the beginning: It should be Seasonal

L362: But isn't March also the peak of the ice season?

L368-370: If the solution is obvious I am curious why it was chosen to not use the dynamical model in NEMO as it can be turned on with a simple namelist switch.

L380: At least in Figure 10 there is no seasonal signal depicted. I think Figure 10 is too cluttered. The mean profile already gives a good impression about the mean state and the stratification. Instead of plotting SD i would appreciate a second Figure where seasonal temperature profiles are shown (JJA, DJF). Maybe this could be a new Figure A5 because as of right now it is impossible to see differences. The quality of the figure should be improved. I also think the markers just make it hard to see the curve.

L384: I would say the immediate layer is not there at all. If I remember correctly this is a hint that the small inflows are not captured correctly. This would also add up to the negative salinity bias.

L393: The evaluation is something the authors do. I would propose: The SST bias fluctuates around zero ...

L395: What is the reason for the shallow depth of the Landsort Deep? I get that you need to probably define a max depth, but why is it not at least the same depth at station BY15 at the Gotland Depth?

L402: It is not shown how momentum fluxes compare in both simulations setups.

L413: What do you mean by too strong prescribed inflow of fresh water? Aren't these observations? Again time series plots of bottom salinity at different stations (BY2, BY5, BY15) could help to see that the problem is related to missing inflow intensities.

L421: The underestimation seems like that the model is roughly 50 percent off at the bottom? Or am I mistaken, maybe I am misinterpreting the colorbars.

L424: I fear that this won't be enough. I think a careful investigation of the conditions at the NS BS interface driving the inflows will be necessary and potentially with subsequent recalibration of vertical and horizontal mixing. May be a slightly longer paragraph here is needed.

Figure 12: I think the figure hints that the inflow dynamics are probably not really well resolved. Again time series plots would help to clarify this. Also the stratifications seem to be underestimated significantly. For future studies processes such as oxygen transportation into the Baltic Sea won't be correctly estimated.

L430: Why only a short period?

Figure 14: I think the colorbar makes it hard to see the differences. At a first glance it looks like that the inflow is not reaching the deeper basins. Would anomalies help here or a different coloring? Maybe you have a better idea.

L582: Freshwater is from observations? How can it be too strong?

L585: I am not sure if the stratification is reproduced as well as the inflows.

Overall, this is a valuable and well-structured contribution, and I appreciate the effort that went into both the coupled setup and the detailed evaluation. Addressing the points above will further strengthen the scientific robustness and clarity of the study.

---

## Author Comment (AC1)

**Answers to the reviewer**

**Review of the manuscript egusphere-2025-3407:**

Evaluation of coupled and uncoupled ocean-ice-atmosphere simulations using icon\_2024.07 and NEMOv4.2.0 for the EURO-CORDEX domain

by Vera Maurer, Wibke Düsterhöft-Wriggers, Rebekka Beddig, Janna Meyer, Claudia Hinrichs, Ha Thi Minh Ho-Hagemann, Joanna Staneva, Birte-Marie Ehlers, and Frank Janssen

The manuscript describes the evaluation of a new atmosphere-ocean coupled model and the two respective atmospheric and oceanic stand-alone models on the ERA5/ORAS5 period, on the EURO-CORDEX domain. The setup of the coupled model is well described. The results are often compared to observations, showing that they are reasonably realistic, and that the coupled model can later be used to perform historic and scenario simulations. The evaluation is more detailed on the oceanic part, which is very interesting per se, but I think it could be a bit enlarged on the atmospheric part.

I find the manuscript well written. The most serious restriction I have about the study concerns the period, which is often inconsistent with the 1979-2021 one mentioned on the first line of the abstract, and the 1979-2020 one on line 216, without any explanation. Indeed the other periods are:

- (1) 1979-2020 on the overview of the experiment (Line 216)
- (2) 1983-2020 for SST from ROAM-NBS except for marine heat waves (MHW from 1989 on)
- (3) 1981-2020 for SST from NEMO-NBS except for MHW (I understand 1981 is chosen for the comparison to Copernicus data, but the 1rst years could be shown at least on fig. A4)
- (4) 1989-2021 for MHW (1989 coinciding with observations)
- (5) 1979-2020 for atmospheric variables
- (6) 1979-2020 in fig. 7 with also the skin temperature
- If (2) is an effect of ocean spin-up (and even 1984 seems out of range in fig. 3 and A4), then the atmospheric variables should not be shown before.
- (4): There is no reason to add 2021 for the MHW only, without showing this year for the other diagnostics.

So to my opinion, one option is to explain why the first years of ROAM-NBS and NEMO-NBS are not shown, and then only show the same period for the atmospheric ROAM-NBS; the second option is to announce a 1983-2020 evaluation simulation.

With this minor revision, I believe the article will be ready for publication.

**Answer:**

Thank you very much for this thorough review and the helpful comments.

The evaluation of the atmosphere over land is only shortly done as the respective atmosphere-only simulations with ICON-CLM are submitted to CORDEX likewise and will be evaluated elsewhere. It is shown here that the atmospheric variables over land are, on average, not strongly affected by the coupling. For the evaluation of the atmospheric part over the ocean area, unfortunately not many high-resolution reference data are available. The seasonal mean precipitation bias against GPM (which also contains data over the ocean) and of longwave radiation against CERES were added to Fig. A1 (also to consider the specific comments below). However, the focus of the evaluation is clearly lying on the ocean part, as stated in the second last sentence of the introduction: "As the ocean component is of additional benefit compared to most other CMIP6-CORDEX simulations, a particular focus is put on the evaluation of NEMO-NBS and the ocean part of ROAM-NBS."

**Figure A1.** Seasonal mean biases of surface shortwave net radiation for ROAM-NBS against CERES (a), of precipitation against E-OBS, of surface longwave net radiation against CERES (c), and of precipitation against IMERG (d), all averaged over 2001–2020.

**On the evaluation period:**

Indeed, the evaluation period of 1979-2020 is not optimal for the ocean part as none of the reference data sets for the ocean is available for the whole period. However, we wanted to stick to the whole period, as it is the minimum period required for CORDEX, for which we will also deliver the data. A more elaborate explanation was added in Sect. 2.4 "Overview of experiments":

"[...] we are overall evaluating and comparing three simulations for the years 1979-2020, which is the minimum period required for CORDEX. However, especially for the ocean part, many reference data are available for shorter time periods only: SST and sea-ice data are available from September 1981, salinity

from 1993 and station data are very sparse before 1993. Therefore, the evaluated time periods had to be adapted in these cases. An overview is given in Tab. 2. For evaluations in which statistics from hourly data were calculated, shorter time periods were selected, partly due to limited data availability, partly to reduce the computational costs."

**Table 2.** Overview of datasets used for evaluation; references are given in the respective sections; the full years were used if not denoted otherwise.

| Dataset                                                                                   | Evaluated variables                          | Used in Sect.    | Evaluated time period                     |
|-------------------------------------------------------------------------------------------|----------------------------------------------|------------------|-------------------------------------------|
| Copernicus ESA SST CCI and C3S
reprocessed SST analyses                                | SST, sea ice                                 | 3.1, 3.3.1, A2   | 1981–2020                                 |
| ERA5                                                                                      | SST                                          | 3.2.1            | 1979–2020                                 |
| E-OBS                                                                                     | tas, tasmin,
tasmax, precipitation        | 3.2.1, A1        | 1979–2020
(2001–2020 in Fig. A1)       |
| meteorological stations                                                                   | hourly wind speed (10 m)                     | 3.2.3            | 2011–2020                                 |
| FINO1 wind measurement                                                                    | hourly wind speed (100 m)                    | 3.2.3            | 2004–2010                                 |
| Copernicus Baltic Sea- In Situ
Near Real Time Observations                             | ocean temperature
and salinity (profiles) | 3.3.2, 3.3.3, A2 | 1979–2020                                 |
| Copernicus Multi Observation Global Ocean
Sea Surface Salinity and Sea Surface Density | surface salinity                             | 3.3.3            | December 1993–November 2020               |
| GESLAv3.0 observational data                                                              | sea surface heights (SSH)                    | 3.3.4, 4.2, A2   | 2015-2019 (or selected events)            |
| Baltic thalweg level 4 dataset                                                            | temperature and
salinity (thalweg)        | 4.1              | November 2014,
February and March 2015 |
| Copernicus Baltic Sea Physics Reanalysis                                                  | SST                                          | 4.3              | 1989-2020                                 |
| CERES                                                                                     | surface radiation                            | A1               | 2001-2020                                 |
| IMERG                                                                                     | precipitation                                | A1               | 2001-2020                                 |
| Copernicus Atlantic- European North West Shelf-
Ocean Physics Reanalysis               | SST                                          | A3               | 1989–2020                                 |

The missing years of ROAM-NBS in Fig. 3 (1981-1982) were due to a post-processing error for SST and not related to a masking out of a spinup effect. The figures 2,3, 10, 12 and A4 were corrected accordingly and the years 1979-1980 were added to the absolute time series in Fig. A4.

Below is the updated version of Fig. 3. We now also consequently masked out points with sea ice in the observations, since the SSTs in the Copernicus reanalysis are artificially set to -1.8°C in the regions covered by sea ice.

**Please find below some more specific remarks, suggestions and corrections: (answers in bold grey)**

- abstract: it should be precised that it is an ERA5/ORAS5 simulation; done
- L12: in the abstract and other parts in the manuscript, the authors say that in the coupled model, the SST bias leads to biases in the ocean-atmosphere fluxes: I don't agree with that. A coupled model develops it's own state of equilibrium, SST and atmosphere-ocean surface are linked.

  The formulation was modified: "Differences of fluxes and precipitation over the ocean between the coupled and uncoupled simulation are largely related to SST differences." See also the answers to the resp. comments below.
- L28: the delay behind global simulations is also true for regional downscaling of the atmosphere, because they are also forced by global simulations;

Exactly this was meant by the "downscaling chain". To make it clearer, the formulation was modified to: "Since the standalone ocean models are ideally forced by the output of the regional atmospheric models, the ocean simulations can only be delivered with a considerable delay compared to the global climate simulations due to the downscaling chain."

- L57: in the manuscript one can find either SI3 or SI3: it should be normalized; done, it was changed to SI3 in the coupling section
- The new CORDEX Task Force on Regional Climate Projections (https://cordex.org/strategic-activities/taskforces/task-force-on-regional-ocean-climate-projections/) could be mentioned in the Introduction. → added in line 27 and 28 within the introduction

- L98: add Northern to "its adjacent seas" (there is no Mediterranean or Black Sea); done
- L137: I think that the tuning mentioned in L262-263 should be presented here
  The values and the switch are given now in the model description section; however, we preferred to keep the main part of the explanation in Sect. 3.2.1 together with the discussion of the results, which is better understandable then
- L138: "For ROAM-NBS as well as for UDAG": add "and ICON-CLM" → done
- L138: change icon-2024.07 for icon\_2024.07 (as in the title and L56)
   changed to icon-2024.07 everywhere to make it consistent with the registered model name on <a href="https://github.com/WCRP-CORDEX/cordex-cmip6-cv">https://github.com/WCRP-CORDEX/cordex-cmip6-cv</a>; the hyphen instead of the underscore is also used on icon-model.org
- L145: two ((; done
- L147: the lateral resolution is 2nm? → changed to: the horizontal resolution is 2 nm
- L148: could the authors precise more the vertical distribution of the 50 layers? Which is the depth of the first ones?

A more detailed description of the vertical distribution being dominantly sigma levels with a hyperbolic tangent transition following Madec et al. 1996, between top and bottom was added in section 2.2. The upper 16 levels are < 1.0m resolution within the whole domain.

- L149: 2 "chosen" → corrected
- L151: the reference is Madec et al. and not Gurvan et al. (to be corrected in the bibliography as well); done
- L182: Craig et al. is the reference for OASIS3-MCT\_3.0: the manual of OASIS3-MCT\_5.0 by Valcke et al. 2021 could be added;  $\rightarrow$  **done**
- L224: is it the same ocean restart for ROAM-NBS?

A sentence was added in the manuscript:

- "The restart field for 1 September 1978 from the spin-up simulation with NEMO-NBS was then used to start ROAM-NBS."
- L245 and followings: replace sea surface temperature by SST; → done
- L247: precise the Copernicus data period;  $\rightarrow$  done
- L253: the authors chose the same seasons as in atmospheric studies, but for information the seasons usually used for ocean variables are JFM, AMJ, JAS, OND;

The chosen "standard" seasons DJF etc allow a better comparability with the atmospheric part.

- L263: cf. L137 above → see above
- L267-268: this parametrisation should be discussed in §2.2 or 2.4;

A section on the turbulence parametrisation including values for eddy-diffusivity and eddy-viscosity is added to 2.2.

- L289: don't put "surface air temperature": it is confusing with surface temperature above; → ok
- Figure 6: the "S" of seasonal is missing; → corrected
- §3.2.2: I think LongWave fluxes should be shown as well; and compare to the observations, not only RAOM-NBS to ICOM-CLM, which can be biased as well;

  Seasonal LW biases against CERES (2001-2020) and precip bias against GPM were added to the appendix (Fig. A3), the text was adapted accordingly; it is a problem to find good measurements (also note that both CERES sfc radiation and GPM precip are derived products and not direct measurements of the respective quantities) over the ocean, especially at appropriate resolutions.
- Fig.7 and in the text: I don't understand how the authors choose now the skin temperature: do they have it as an output of NEMO4 (and from 1979?)? The SST comparison to ERA5 SST (which is the one imposed to ICON-CLM) is enough, as it was compared to Copernicus before. Thank you for the hint, the nomenclature here was indeed a bit confusing. The reason is that we tried to stick to cmor variable names (tas, tasmin, tasmax, skt, lhfl, ...) where applicable. In this part of the evaluation, skt refers to ICON output (T\_G = surface temperature) which is (over the ice-free ocean) identical to the SST, as we are not using a dedicated skin temperature parameterization over the ocean as e.g. IFS (it is available in ICON now, but not yet in the version 2024.07 which we are using). It is the same quantity as shown in Fig. 4. We now replaced all occurrences of skin temperature with surface temperature (and skt with Tsfc).
- L316: I don't like the formulation "the skt difference determines the sign of the flux..."; as I said before they are linked in the coupled model; besides they qualify their statements on L325; The main aim of 3.2.2 is to show that there is a strong relationship between the SST biases (which are not identical, but still very similar in the coupled and uncoupled ocean parts, which was discussed in the first parts of Sect. 3) and the flux differences. As the SST in ICON-CLM is prescribed by ERA5, it is assumed to be more realistic than in ROAM-NBS and therefore, the SST difference between both is sometimes called "bias". The interpretation is that the precipitation and flux differences between the coupled and the uncoupled ICON-CLM can be largely explained by the SST differences. The resp. text passages were re-formulated and an introduction was added to Sect 3.2.2, which explains why the evaluation is done in that way.
- L389: "which" will be shown; done
- L393: a space is missing between "evaluation," and "the bias"; done
- L413-414: indeed the sea surface salinity is highly linked to the E-P-R flux; concerning the runoff, there are options in NEMO which for example propagate the runoff through the vertical, and also to enhance the vertical mixing at the river mouths: the choice here could be precised in § 2.2 or 2.4, or it could be discussed here;

An evaluation for sea surface salinity was done for both NEMO-NBS and ROAM-NBS and is only shown for ROAM-NBS due to minimal differences in the bias, therefore the E-P-R flux was not identified as the main reason for the bias. A one year test run using the ehype runoff data instead of the presented mix of observational and WaterGap runoff data showed promising results of an

approximately 1 psu smaller salinity biases along German coasts. Within the current NEMO-NBS and ROAM-NBS setups, the runoff is only applied in the upper layer and no enhanced treatment available in NEMO is applied. A sentence discussing these options was added in L413-414.

- L425: the restart of the simulation can sometimes also explain some biases in the deeper layers; Thank you for this comment, that is of course correct, especially within the enclosed basins in the Baltic Sea. A small addition referencing the initial data was added in L425.
- L433: add fig.8 for Cuxhaven;

A reference to Fig. 8 was added after mentioning the station Cuxhaven.

- L435: add "nearly" at "a higher correlation at all station" because it's not the case for all; done
- L444: I'm not a specialist, but is "wind surge" appropriate here when the authors show only the SSH? Thank you for this attentive remark. Indeed the term wind surge is not quite appropriate here and was changed to storm surge throughout the complete document. In the SSH evaluation section mainly the detided SSH results with additionally removed mean sea level are presented in the figures as well as the table and therefore the storm surge is evaluated as explained in lines 431-433. The explanation is updated to a more detailed version and the text in this section is slightly adjusted for more clarity.
- Fig. 13: replace "blue" by "left" and "green" by "right";
  Thank you for the remark, this was included and the text in brackets changed.
- Fig. 14: what are the isolines for?

The isolines display discrete values of salinity and temperature for easier comparison of the stratification within the Baltic. In the caption of Fig. 14 and Fig. 15 a descriptive sentence was added.

- L465: it would be of interest to explain what is the "normal" situation of the inflow in the Baltic; A section on the "normal" inflow situation in the Baltic was added in 4.1, including three new literature citations and a short definition of Major Baltic Inflow events.
- Fig. 16 c: add ICON-CLM?

ERA5 is included instead of ICON-CLM as this is used for the forcing of NEMO-NBS (both Fig. 16 c show then consistently observations, results/forcing from NEMO-NBS and results from ROAM-NBS)

- L517: I guess Lighthouse Kiel is also Leuchtturm Kiel of fig.8: choose the same name; The station name was changed to Leuchtturm Kiel for consistency.
- Fig. 17: Orange stars, not green; it would be interesting to show the MHW from the beginning of the simulations, even if there are no observations;

Corrected, thank you for pointing this out. MHW were chosen not to be shown from 1979 due to missing observational data.

- L523: idem L517, and add Cuxhaven of fig. 8 for UFS German Bight (if I'm right?)

That is the station UFS Deutsche Bucht. We changed the station name here to German. Cuxhaven is a different station.

- L536: the authors must compute the correlations to say that;
  The computed correlation based on linear regression for both models to observed MHW frequency, intensity and days at station Leuchtturm Kiel has been added to the text.
- Chapter 4: at the end of this chapter it would be nice to add a conclusion; →added: "Overall, the evaluation of variability and extreme events shows that both NEMO-NBS and ROAM-NBS can generally reproduce but underestimate the Major Baltic inflow event, that they are able to represent storm surge events, and capture MHWs."
- L582: the imposed runoff comes from observations, so do the authors think they might be too strong? cf. Remark for L413-414, and also there could be a discussion about coupling the runoff with a hydrological model: it is the best solution for future scenario simulations. Besides the E-P budget is also of much importance in the surface salinity;

Yes, since our test run using a different runoff data set (ehype) results in an approximately 1psu smaller salinity bias along German coasts than our evaluation runs using the mixed observational and model data set provided by BfG, we think that the runoff combined with prescribing it in only the upper cell is too strong, also cf. comment above (L413-414). Yes, for the coupled historicals and scenarios, the online coupling with HD will be used. A comment was added in the text.

- L593: indeed the spin-up period is very important for the ocean, but also for the coupled model to reach its equilibrium, and as I said before concerning the period of the study, the authors must explain if they think that the 1rst years of the ROAM-NBS simulation is in a spin-up phase.

  After correcting Fig. 2 and Fig A4, it is much clearer that the coupled model does not show an additional spinup phase. Especially the absolute time series show this.
- L607: the authors don't show the computed correlations;
  The computed correlation based on linear regression for both models to observed MHW frequency, intensity and days at station Leuchtturm Kiel has been added to the text in the MHW chapter.
- L633: add Fig. Before 6a → done
- Fig. A5: replace "left, mid, right" by "a, b, c" → done
- Fig. A7: replace "blue" by "left" and "green" by "right" → done

---

## Author Response (AR1)

**Author's response**

**Answers to reviewer 1**

**Review of the manuscript egusphere-2025-3407:**
Evaluation of coupled and uncoupled ocean-ice-atmosphere simulations using icon_2024.07 and NEMOv4.2.0 for the EURO-CORDEX domain by Vera Maurer, Wibke Düsterhöft-Wriggers, Rebekka Beddig, Janna Meyer, Claudia Hinrichs, Ha Thi Minh Ho-Hagemann, Joanna Staneva, Birte-Marie Ehlers, and Frank Janssen

The manuscript describes the evaluation of a new atmosphere-ocean coupled model and the two respective atmospheric and oceanic stand-alone models on the ERA5/ORAS5 period, on the EURO-CORDEX domain. The setup of the coupled model is well described. The results are often compared to observations, showing that they are reasonably realistic, and that the coupled model can later be used to perform historic and scenario simulations. The evaluation is more detailed on the oceanic part, which is very interesting per se, but I think it could be a bit enlarged on the atmospheric part.
I find the manuscript well written. The most serious restriction I have about the study concerns the period, which is often inconsistent with the 1979-2021 one mentioned on the first line of the abstract, and the 1979-2020 one on line 216, without any explanation. Indeed the other periods are:
- (1) 1979-2020 on the overview of the experiment (Line 216)
- (2) 1983-2020 for SST from ROAM-NBS except for marine heat waves (MHW from 1989 on)
- (3) 1981-2020 for SST from NEMO-NBS except for MHW (I understand 1981 is chosen for the comparison to Copernicus data, but the 1rst years could be shown at least on fig. A4)
- (4) 1989-2021 for MHW (1989 coinciding with observations)
- (5) 1979-2020 for atmospheric variables
- (6) 1979-2020 in fig. 7 with also the skin temperature
If (2) is an effect of ocean spin-up (and even 1984 seems out of range in fig. 3 and A4), then the atmospheric variables should not be shown before.
(4): There is no reason to add 2021 for the MHW only, without showing this year for the other diagnostics.

So to my opinion, one option is to explain why the first years of ROAM-NBS and NEMO-NBS are not shown, and then only show the same period for the atmospheric ROAM-NBS; the second option is to announce a 1983-2020 evaluation simulation.
With this minor revision, I believe the article will be ready for publication.

**Answer (general comments)**

Thank you very much for this thorough review and the helpful comments.

[Figure]

***Figure A1.*** Seasonal mean biases of surface shortwave net radiation for ROAM-NBS against CERES (a), of precipitation against E-OBS, of surface longwave net radiation against CERES (c), and of precipitation against IMERG (d), all averaged over 2001–2020.

The evaluation of the atmosphere over land is only shortly done as the respective atmosphere-only simulations with ICON-CLM are submitted to CORDEX likewise and will be evaluated elsewhere. It is shown here that the atmospheric variables over land are, on

average, not strongly affected by the coupling. For the evaluation of the atmospheric part over the ocean area, unfortunately not many high-resolution reference data are available. The seasonal mean precipitation bias against GPM (which also contains data over the ocean) and of longwave radiation against CERES were added to Fig. A1 (also to consider the specific comments below). However, the focus of the evaluation is clearly lying on the ocean part, as stated in the second last sentence of the introduction: "As the ocean component is of additional benefit compared to most other CORDEX-CMIP6 simulations, a particular focus is put on the evaluation of NEMO-NBS and the ocean part of ROAM-NBS."

On the evaluation period:
Indeed, the evaluation period of 1979-2020 is not optimal for the ocean part as none of the reference data sets for the ocean is available for the whole period. However, we wanted to stick to the whole period, as it is the minimum period required for CORDEX, for which we will also deliver the data.
A more elaborate explanation was added in Sect. 2.4 "Overview of experiments and availability of reference data":
"The evaluation is generally conducted for the years 1979-2020, which is the minimum period required for CORDEX. However, especially for the ocean part, many reference data are available for shorter time periods only, so that the evaluation period had to be adapted in these cases: SST and sea ice data (Copernicus ESA SST CCI and C3S reprocessed SST analyses; https://doi.org/10.48670/moi-00169) are available from September 1981 and surface salinity from 1993 (Copernicus Multi Observation Global Ocean Sea Surface Salinity and Sea Surface Density; https://doi.org/10.48670/moi-00051). Station data (Copernicus Baltic Sea- In Situ Near Real Time Observations; https://doi.org/10.48670/moi-00032) are very sparse before 1993. Copernicus reanalyses used for the evaluation of marine heat waves (Copernicus Baltic Sea Physics Reanalysis; https://doi.org/10.48670/moi-00013; Atlantic- European North West Shelf- Ocean Physics Reanalysis; https://doi.org/10.48670/moi-00059) start in 1989. For the atmospheric part, satellite data used for the evaluation of surface radiation (CERES; NASA/LARC/SD/ASDC, 2019) are available from 2001 only. Therefore, the evaluated time periods had to be adapted in these cases; an overview is given in Tab. 2. For evaluations in which statistics were calculated from hourly data, shorter time periods were used, partly due to limited data availability as well, partly to reduce the computational costs (see Sects. 3.2.3 and 3.3.4)."

**Table 2.** Overview of datasets used for evaluation; references are given in the text; the full years were used if not denoted otherwise.

| Dataset | Evaluated quantities | Used in Sect. | Evaluated time period |
|---|---|---|---|
| Copernicus ESA SST CCI and C3S reprocessed SST analyses | SST, sea ice | 3.1, 3.3.1, A2 | 1981–2020 |
| ERA5 | SST | 3.2.1 | 1979–2020 |
| E-OBS | tas, tasmin, tasmax, precipitation | 3.2.1, A1 | 1979–2020; 2001–2020 in Fig. A1 |
| meteorological stations | hourly wind speed (10 m) | 3.2.3 | 2011–2020 |
| FINO1 wind measurement | hourly wind speed (100 m) | 3.2.3 | 2004–2010 |
| Copernicus Baltic Sea- In Situ Near Real Time Observations | ocean temperature and salinity (profiles) | 3.3.2, 3.3.3, A2 | 1979–2020 |
| Copernicus Multi Observation Global Ocean Sea Surface Salinity and Sea Surface Density | surface salinity | 3.3.3 | December 1993–November 2020 |
| GESLAv3.0 observational data | sea surface heights (SSH) | 3.3.4, 4.2, A2 | 2015–2019 (or selected events) |
| Baltic thalweg level 4 dataset | temperature and salinity (thalweg) | 4.1 | November 2014, February and March 2015 |
| Copernicus Baltic Sea Physics Reanalysis | SST | 4.3 | 1989–2020 |
| CERES | surface radiation | A1 | 2001–2020 |
| IMERG | precipitation | A1 | 2001–2020 |
| Copernicus Atlantic- European North West Shelf- Ocean Physics Reanalysis | SST | A3 | 1989–2020 |

The missing years of ROAM-NBS in Fig. 3 (1981-1982) were due to a post-processing error for SST and not related to a masking out of a spinup effect. The figures 2, 3, and A4 were corrected accordingly and the years 1979-1980 were added to the absolute time series in Fig. A4.

Below is the updated version of Fig. 3. We now also consequently masked out points with sea ice in the observations, since the SSTs in the Copernicus reanalysis are artificially set to -1.8°C in the regions covered by sea ice.

[Figure]

**Figure 3:** Bias of winter (DJF) and summer (JJA) seasonal mean SST time series for NEMO-NBS and ROAM-NBS to Copernicus observation data for the whole NEMO-NBS domain (a), Open Atlantic (b), the North Sea (c) and the Baltic Sea (d).

**Answers (specific comments)**

**Please find below some more specific remarks, suggestions and corrections:**
**(answers in bold grey)**

- abstract: it should be precised that it is an ERA5/ORAS5 simulation; **done**

- L12: in the abstract and other parts in the manuscript, the authors say that in the coupled model, the SST bias leads to biases in the ocean-atmosphere fluxes: I don't agree with that. A coupled model develops it's own state of equilibrium, SST and atmosphere-ocean surface are linked.
**The formulation was modified:**
"Differences in fluxes and precipitation over the ocean between the coupled and uncoupled simulation are largely related to SST differences."
**See also the answers to the respective comments below.**

- L28: the delay behind global simulations is also true for regional downscaling of the atmosphere, because they are also forced by global simulations;
**Exactly this was meant by the "downscaling chain". To make it clearer, the formulation was modified:**
"Since the standalone ocean models are ideally forced by the output of the regional atmospheric models, the ocean simulations can only be delivered with a considerable delay compared to the global climate simulations due to the downscaling chain."

- L57: in the manuscript one can find either SI3 or SI^3: it should be normalized;

**done, it was changed to SI3 in the coupling section**

- The new CORDEX Task Force on Regional Climate Projections
(https://cordex.org/strategic-
activities/taskforces/task-force-on-regional-ocean-climate-projections/) could be mentioned
in the Introduction.
**We added a sentence in the Introduction:**
"Only in 2025, the CORDEX Task Force on Regional Ocean  Climate Projections
(https://cordex.org/strategic-activities/taskforces/task-force-on-regional-ocean-climate-project
ions; last access: 25 September 2025) was established."

- L98: add Northern to "its adjacent seas" (there is no Mediterranean or Black Sea); **done**

- L137: I think that the tuning mentioned in L262-263 should be presented here
**A short explanation of the tuning parameters *allow_overcast* (which was mentioned in
L262-263, Sect. 3.2.1 in the submitted version) and *tkhmin* (Sect. 3.2.2) is now given in
the model description section:**
"For ICON-CLM as well as for ROAM-NBS, an important tuning target was the reduction of a
positive surface shortwave radiation bias over land and ocean. In the current version, a
tuning of the cloud cover scheme (monthly varying *allow_overcast* parameter) is applied.
One setting that was not modified in ICON-CLM-UDAG compared to NWP, but in our
ICON-CLM and ROAM-NBS simulations, was the minimum diffusion coefficient for heat
*tkhmin*. It was increased from 0.6 to 0.8 for the winter months December through February.
The effects of this modified setting will be discussed in Sect. 3.2.1."

- L138: "For ROAM-NBS as well as for UDAG" : add "and ICON-CLM" → **done**

- L138: change icon-2024.07 for icon_2024.07 (as in the title and L56)
**Changed to icon-2024.07 everywhere to make it consistent with the registered model
name on https://github.com/WCRP-CORDEX/cordex-cmip6-cv ; the hyphen instead of
the underscore is also used on icon-model.org**

- L145: two ((; **done**

- L147: the lateral resolution is 2nm?   → **changed to: the horizontal resolution is 2 nm**

- L148: could the authors precise more the vertical distribution of the 50 layers? Which is the
depth of
the first ones?
**A more detailed description of the vertical distribution being dominantly sigma levels
with a hyperbolic tangent transition following Madec et al. 1996, between top and
bottom was added in section 2.2. The upper 16 levels are < 1.0m resolution within the
whole domain. Reviewer 2 had a related comment, so that the description was also
extended:**
"This vertical coordinate allows for the representation of the deeper and shallower regions
simultaneously and consists of a predominantly σ-coordinate with a hyperbolic transient
transition between the top and bottom layers following Madec and Imbard (1996).

While creating the domain file containing the smoothed bathymetry used by NEMO, a slope is determined at which the terrain-following σ-coordinate intersects the sea bed and becomes a pseudo z-coordinate. This transition leads to a smaller bottom-level index in areas with steep slopes, like the North West Shelf.

A high resolution in the upper layers (< 0.18 m in the top layer and < 1.0 m within the upper 16 layers) leads to a good representation of the ocean's interface to the atmosphere."

- L149: 2 "chosen" → **corrected**

- L151: the reference is Madec et al. and not Gurvan et al. (to be corrected in the bibliography as well); **done**

- L182: Craig et al. is the reference for OASIS3-MCT_3.0: the manual of OASIS3-MCT_5.0 by Valcke et al. 2021 could be added; → **done**

- L224: is it the same ocean restart for ROAM-NBS?
**A sentence was added in the manuscript:**
"The restart field for 1 September 1978 from the spin-up simulation with NEMO-NBS was then used to start ROAM-NBS."

- L245 and followings: replace sea surface temperature by SST; → **done**

- L247: precise the Copernicus data period; → **done**

- L253: the authors chose the same seasons as in atmospheric studies, but for information the seasons usually used for ocean variables are JFM, AMJ, JAS, OND;
**The chosen "standard" seasons DJF etc allow a better comparability with the atmospheric part.**

- L263: cf. L137 above → **see above**

- L267-268: this parametrisation should be discussed in §2.2 or 2.4;
**A section on the turbulence parametrisation including values for eddy-diffusivity and eddy-viscosity is added to 2.2.**

- L289: don't put "surface air temperature": it is confusing with surface temperature above; → **ok, deleted**

- Figure 6: the "S" of seasonal is missing; → **corrected**

- §3.2.2: I think LongWave fluxes should be shown as well; and compare to the observations, not only
RAOM-NBS to ICOM-CLM, which can be biased as well;
**Seasonal LW biases against CERES (2001-2020) and precip bias against GPM were added to the appendix (Fig. A3), the text was adapted accordingly; it is a problem to find good measurements (also note that both CERES sfc radiation and GPM precip are derived products and not direct measurements of the respective quantities) over the ocean, especially at appropriate resolutions.**

- Fig.7 and in the text: I don't understand how the authors choose now the skin temperature: do they have it as an output of NEMO4 (and from 1979?)? The SST comparison to ERA5 SST (which is the one imposed to ICON-CLM) is enough, as it was compared to Copernicus before.

**Thank you for the hint, the nomenclature here was indeed a bit confusing. The reason is that we tried to stick to cmor variable names (tas, tasmin, tasmax, skt, lhfl, …) where applicable. In this part of the evaluation, skt refers to ICON output (T_G = surface temperature) which is (over the ice-free ocean) identical to the SST, as we are not using a dedicated skin temperature parameterization over the ocean as e.g. IFS (it is available in ICON now, but not yet in the version 2024.07 which we are using). It is the same quantity as shown in Fig. 4. We now replaced all occurrences of skin temperature with surface temperature (and skt with Tsfc).**

- L316: I don't like the formulation "the skt difference determines the sign of the flux…"; as I said before they are linked in the coupled model; besides they qualify their statements on L325;

**The main aim of 3.2.2 is to show that there is a strong relationship between the SST biases (which are not identical, but still very similar in the coupled and uncoupled ocean parts, which was discussed in the first parts of Sect. 3) and the flux differences. As the SST in ICON-CLM is prescribed by ERA5, it is assumed to be more realistic than in ROAM-NBS and therefore, the SST difference between both is sometimes called "bias". The interpretation is that the precipitation and flux differences between the coupled and the uncoupled ICON-CLM can be largely explained by the SST differences. The respective text passages were re-formulated and an introduction was added to Sect 3.2.2, which explains why the evaluation is done in that way:**

"To give an overview of precipitation and fluxes over the ocean, for which good measurement products are not available or are not at a sufficient spatial resolution to adequately evaluate the NBS region, the differences between the coupled and uncoupled simulations are analyzed here. For completeness, seasonal mean bias maps are provided for precipitation and surface net longwave radiation in Fig. A1 in the Appendix. As the SST is prescribed for ICON-CLM, the SST differences between ROAM-NBS and ICON-CLM also reflect the bias against observations, which does not directly mean that the precipitation and flux differences between both simulations, as shown here, reflect biases. However, they can give a good indication of the reaction of the coupled model to SST biases."

- L389: "which" will be shown; **done**

- L393: a space is missing between "evaluation," and "the bias"; **done**

- L413-414: indeed the sea surface salinity is highly linked to the E-P-R flux; concerning the runoff, there are options in NEMO which for example propagate the runoff through the vertical, and also to enhance the vertical mixing at the river mouths: the choice here could be precised in § 2.2 or 2.4, or it could be discussed here;

**An evaluation for sea surface salinity was done for both NEMO-NBS and ROAM-NBS and is only shown for ROAM-NBS due to minimal differences in the bias, therefore the E-P-R flux was not identified as the main reason for the bias. A one year test run using the ehype runoff data instead of the presented mix of observational and**

**WaterGap runoff data showed promising results of an approximately 1 psu smaller salinity biases along German coasts. Within the current NEMO-NBS and ROAM-NBS setups, the runoff is only applied in the upper layer and no enhanced treatment available in NEMO is applied. A sentence discussing these options was added in Sect. 3.3.3:**

"Within the current NEMO-NBS and ROAM-NBS simulations, the runoff is prescribed only in the upper layer. A prescription over the complete water column and additional vertical mixing could better distribute the fresh water from the rivers and further enhance the sea surface salinity results."

- L425: the restart of the simulation can sometimes also explain some biases in the deeper layers;
**Thank you for this comment, that is of course correct, especially within the enclosed basins in the Baltic Sea. A small addition referencing the initial data was added at the end of Sect. 3.3.3 together with improved mixing according to a comment by reviewer 2:**

"Overall, an underestimation of bottom salinity in deeper Baltic basins can be observed. This underestimation could be mitigated in future simulations by employing longer spin-up periods or improved initial conditions. Additionally, efforts should focus on refining vertical and horizontal mixing within the Baltic and increasing spatial resolution in the Danish Straits."

- L433: add fig.8 for Cuxhaven;
**A reference to Fig. 8 was added after mentioning the station Cuxhaven.**

- L435: add "nearly" at "a higher correlation at all station" because it's not the case for all;
**Added in Sect. 3.3.4:**
"For detided sea surface heights, NEMO-NBS shows a higher correlation with the observational data than ROAM-NBS at **nearly** all stations, with minimal differences in correlation coefficients."

- L444: I'm not a specialist, but is "wind surge" appropriate here when the authors show only the SSH?
**Thank you for this attentive remark. Indeed the term wind surge is not quite appropriate here and was changed to storm surge throughout the complete document. In the SSH evaluation section mainly the detided SSH results with additionally removed mean sea level are presented in the figures as well as in the table and therefore the storm surge is evaluated as explained in lines 431-433. The explanation is updated to a more detailed version and the text in Sect. 3.3.4 is slightly adjusted for more clarity:**

"The storm surge component of the water level is assessed by removing the influence of astronomical tides from the bias-adjusted sea surface height. This is achieved using a Demerliac filter, which separates tidal fluctuations from the total signal. The remaining residual captures the non-tidal water level variations caused by meteorological forcing, providing an estimate of the storm surge."

- Fig. 13: replace "blue" by "left" and "green" by "right";
**Thank you for the remark, this was included and the text in brackets changed.**

- Fig. 14: what are the isolines for?

**The isolines displayed discrete values of salinity and temperature for easier comparison of the stratification within the Baltic. An update of the figures was also requested by reviewer 2. In the new versions of Figs. 14 and 15, the isolines were removed and a discrete colorbar introduced.**

- L465: it would be of interest to explain what is the "normal" situation of the inflow in the Baltic;

**A section on the "normal" inflow situation in the Baltic was added in Sect. 4.1, including three new literature citations and a short definition of Major Baltic Inflow events:**

"Under typical conditions, the inflow from the North Sea into the Baltic Sea through the Danish Straits is relatively moderate and exhibits a seasonal cycle. Inflow generally occurs as dense, saline water entering the Baltic and is strongest during late winter and early spring due to wind-driven and barotropic forcing (Matthäus and Franck, 1992; Mohrholz, 2018). These inflows gradually ventilate the deep basins of the Baltic and maintain the salinity balance (Lehmann et al., 2022). Under normal circumstances, the inflow is steady and predictable, with variations primarily controlled by seasonal winds, sea-level differences between the North Sea and the Baltic Sea, and long-term atmospheric pressure patterns (Mohrholz, 2018). Significant episodic events in which dense, saline water from the North Sea flows through the Danish Straits into the Baltic Sea, ventilating its deep basins and temporarily raising deep-water salinity and oxygen levels are called MBI events."

- Fig. 16 c: add ICON-CLM?

**ERA5 is included instead of ICON-CLM as this is used as forcing of NEMO-NBS (with that, both Figs. 16c and 16d consistently show observations, results/forcing from NEMO-NBS and results from ROAM-NBS).**

- L517: I guess Lighthouse Kiel is also Leuchtturm Kiel of fig.8: choose the same name;

**The station name was changed to "Leuchtturm Kiel" in Sect. 4.3 for consistency. The reference to Fig. 8 was added as well.**

- Fig. 17: Orange stars, not green; it would be interesting to show the MHW from the beginning of the simulations, even if there are no observations;

**Corrected, thank you for pointing this out. MHW were chosen not to be shown from 1979 due to missing observational data.**

- L523: idem L517, and add Cuxhaven of fig. 8 for UFS German Bight (if I'm right?)

**That is the station "UFS Deutsche Bucht". We changed the station name in Sect. 4.3 to German. Cuxhaven is a different station.**

- L536: the authors must compute the correlations to say that;

**The computed correlation based on linear regression for both models to observed MHW frequency, intensity and days at station Leuchtturm Kiel has been added in Sect. 4.3.**

- Chapter 4: at the end of this chapter it would be nice to add a conclusion;

**We added:**

"Overall, the evaluation of variability and extreme events shows that both NEMO-NBS and ROAM-NBS can generally reproduce but underestimate the Major Baltic inflow event, that they are able to represent storm surge events, and capture MHWs."

- L582: the imposed runoff comes from observations, so do the authors think they might be too strong? cf. Remark for L413-414, and also there could be a discussion about coupling the runoff with a hydrological model: it is the best solution for future scenario simulations. Besides the E-P budget is also of much importance in the surface salinity;

**Yes, we think the prescribed runoff is too strong, since our test run using a different runoff data set (E-HYPE) results in an approximately 1 psu smaller salinity bias along German coasts than our evaluation runs using the mixed observational and model data set provided by BfG. We also suppose that prescribing the runoff in the upper cell only contributes (also cf. comment above on L413-414).**

**To the solution for future scenario simulations: Yes, for the coupled historicals and scenarios, the online coupling with HD will be used. A comment was added in the text (Conclusions):**

"These [salinity] biases may be attributed to overly strong prescribed freshwater runoff, which is not exclusively observation-based, and insufficient representation of saline inflow from the North Sea. Prescribing runoff throughout the water column and enhancing vertical mixing could improve sea surface salinity in NEMO-NBS and ROAM-NBS simulations.

For the generation of the historical simulations, an online coupled runoff model will be used as in Hagemann et al. (2024), which is already available in the setup but was not used for better comparability between the coupled and the ocean-only simulation."

- L593: indeed the spin-up period is very important for the ocean, but also for the coupled model to reach its equilibrium, and as I said before concerning the period of the study, the authors must explain if they think that the 1rst years of the ROAM-NBS simulation is in a spin-up phase.

**After correcting Fig. 2 and Fig A4, it is much clearer that the coupled model does not show an additional spinup phase. Especially the absolute time series can show this.**

- L607: the authors don't show the computed correlations;

**The computed correlation based on linear regression for both models to observed MHW frequency, intensity and days at station Leuchtturm Kiel has been added to the text in Sect. "4.3 Marine heatwaves":**

**"At Leuchtturm Kiel, the NEMO-NBS simulation has a slightly higher Pearson correlation coefficient r with the observed events (NEMO-NBS: 0.85, ROAM-NBS: 0.84), intensity (NEMO-NBS: 0.57, ROAM-NBS: 0.30, i.e. not significant) and MHW days (NEMO-NBS: 0.97, ROAM-NBS: 0.92)."**

- L633: add Fig. Before 6a → **the text has changed as also Fig. A6 was updated as suggested by reviewer 2 (salinity timeseries for 2 different levels instead of Hovmöller plots are now shown).**

- Fig. A5: replace "left, mid, right" by "a, b, c" → **done**

- Fig. A7: replace "blue" by "left" and "green" by "right" → **done**

**Answers to reviewer 2**

The preprint represents an evaluation of ROAM-NBS (ICON + NEMO v4.2 + SI3, coupled via OASIS3-MCT, alongside the uncoupled components for the period 1979 - 2020. The mean climate in the atmosphere is generally well reproduced with remaining issues concentrating on an observed SST bias, sea ice overestimation in the Baltic Sea and too low salinities in the deeper basins in the Baltic Sea impacting stratification and inflow representation. The study itself is timely and highly relevant.
The paper is well written. Find suggested revisions below:

**Major concerns:**
In the beginning of the paper you stress that the availability of regional ocean projections is sparse and emphasize that the focus of the evaluation will be on the Baltic Sea region.

However, I find that the representation of the Baltic Sea dynamics is not yet satisfactory. In my view, there are two possible ways forward:
1. Recalibrate the model - which I know is a long and stressful approach.
2. If the model cannot yet capture key dynamics (beyond surface variables) in the Baltic Sea, I suggest explicitly stating that the system is not yet production-ready in that regard and that further improvement is needed.

Alternatively, if I am mistaken, I welcome further clarification. My intent is not to criticize the effort but to ensure the results are accurately contextualized. This is in no way an offense to the great work that is being presented here.

**Answer (general comments)**

Dear reviewer, thank you very much for your helpful comments.

We have put a certain emphasis on the evaluations in the Baltic Sea, as this is the area where the quality of the current NEMO setup is most critical.
However, the main research area is not the Baltic Sea. As stated in the abstract, "our target area [...] are the German national waters", including the German North Sea and Baltic Coasts. Since the model encompasses a larger area, we are demonstrating that the mean state and extremes of the whole NBS area are adequately represented so that the results may be used for the mentioned main area of interest.
We make it clearer now that there must be more improvements in the Baltic Sea by modifying the last sentence of the abstract and adding a new one (omitting some other

details before to keep it sufficiently concise):
"Overall, the coupled simulation demonstrates **adequate** performance for both the atmosphere and the ocean, and the setup will be used to produce coupled regional climate projections for Europe. **However, bias correction for the deeper Baltic layers remains necessary for further applications, and future work will focus on refining the setup for this region.**"
With that, we omit the phrase "ready for production" in the revised version and replaced "good performance" with "adequate performance".
Also in the conclusions, we added the sentence "For further applications, it is noteworthy that biases prevail particularly in the Baltic Sea."

However, we cannot re-calibrate the system at the moment, otherwise we would be too late for CORDEX-CMIP6. And this article is meant to describe the quality of the evaluation simulation, which is a requirement for CORDEX, in a transparent way.
Additionally, the production simulations may teach us further lessons - we do not know yet about the performance of all the historical simulations, they might have different biases again. Of course, we will address the biases revealed by this quite broad analysis of our evaluation simulation in the future and apply further calibration. Different options are mentioned in the conclusions and at other locations in the text (see answers to the specific comments).

Ultimately, we can show that the Baltic inflows are qualitatively simulated, even if they are quantitatively underestimated. The salinity time series that you have suggested and which we have added in the appendix of the new manuscript's version (see the Figure below in the answers to the specific comments) confirm that.

**Answers (specific comments)**

**I will try to explain my concerns below along with the text.**
**See answers in bold grey.**
Line references:

L148: Please clarify the implementation of the σ–z* grid. At which depth or criterion switches the model between σ and z coordinates.

**The setup uses a pseudo σ–z\* grid. This is the setting in the model's main running namelist. However, during the creation process of the grid file, specifications for the stretching of the layers had to be done, which implement the pseudo z-coordinate at the bottom besides the σ–coordinate above.**
**The original sentence in Sect. 2.2 was modified:**
"This vertical coordinate allows for the representation of the deeper and shallower regions simultaneously and consists of a predominantly σ-coordinate with a hyperbolic transient transition between the top and bottom layers following Madec and Imbard (1996). While creating the grid file specifically used by NEMO with a smoothed bathymetry, a slope is

determined at which the terrain-following σ-coordinate intersects the sea bed and becomes a pseudo z-coordinate. This transition leads to a smaller bottom-level index in the areas with steep slopes like the North West Shelf."

L191: 2x respective → **done**

L197: Please be consistent with SI^3 or SI3 → **done, now SI3**

L199: How is the albedo over the water set?

**A short explanation was added (Sect. 2.3):**
"Over the ocean, ICON uses a formulation by Taylor et al. (1996) for the direct albedo and the value of 0.06 for diffuse albedo as in ECMWF's IFS model. Additionally, an albedo increase by whitecap cover after Séférian et al. (2018) is considered."

Table 1: Why is rain bold?

**The whole formula for EMP is given to denote how precipitation is used in the coupling interface of NEMO. An explanation was added in the Table caption:** "Variables that are combined into other quantities directly in the coupling interface are written in bold font"

L218: Maybe some overview figure/table addressing the different evaluation periods would be good. I lose track throughout the article.

**We apologize for the confusion. This point was also raised by the other reviewer. We added an overview table (new Tab. 2, see p. 4 of this document) and a paragraph in Sect. 2.4 "Overview of experiments and availability of reference data":**
"The evaluation is generally conducted for the years 1979-2020, which is the minimum period required for CORDEX. However, especially for the ocean part, many reference data are available for shorter time periods only, so that the evaluation period had to be adapted in these cases: SST and sea ice data (Copernicus ESA SST CCI and C3S reprocessed SST analyses; https://doi.org/10.48670/moi-00169) are available from September 1981 and surface salinity from 1993 (Copernicus Multi Observation Global Ocean Sea Surface Salinity and Sea Surface Density; https://doi.org/10.48670/moi-00051). Station data (Copernicus Baltic Sea- In Situ Near Real Time Observations; https://doi.org/10.48670/moi-00032) are very sparse before 1993. Copernicus reanalyses used for the evaluation of marine heat waves (Copernicus Baltic Sea Physics Reanalysis; https://doi.org/10.48670/moi-00013; Atlantic- European North West Shelf- Ocean Physics Reanalysis; https://doi.org/10.48670/moi-00059) start in 1989. For the atmospheric part, satellite data used for the evaluation of surface radiation (CERES; NASA/LARC/SD/ASDC, 2019) are available from 2001 only. Therefore, the evaluated time periods had to be adapted in these cases; an overview is given in Tab. 2. For evaluations in which statistics were calculated from hourly data, shorter time periods were used, partly due to limited data availability as well, partly to reduce the computational costs (see Sects. 3.2.3 and 3.3.4)."

L222: Is 4 years of spin up really enough for the Baltic Sea. I would be really interested to see the timeseries of stations BY2, BY5, BY15 for surface salinity and bottom salinity (similar to Hordoir et al., 2019 their Figure 8).

**Thank you for suggesting the salinity time series figures at stations BY2, BY5, and BY15. They are presented in the Appendix in the new version of the manuscript for both surface and bottom layers. The simulated surface salinity agrees well with observational data, while the interannual variability of the bottom salinity is generally well captured, albeit with a consistent background bias. Major inflow events in 1993, 2003, and 2015 are also reasonably well reproduced. We agree that the ocean model employed a relatively short spin-up period, as acknowledged by the authors in the conclusions (line 593 of the submitted version). This limitation will be addressed in future production runs to improve model performance.**
**To avoid duplication of information and save figure space, the salinity hovmoeller diagrams are taken out of the manuscript and are replaced with the described time series.**

[Figure]

***Figure A6:*** Surface and bottom time series of observed and simulated salinity at Baltic stations (SMHIBY2, SMHIBY5, SMHIBY15).

L234: The sentence is not so easy to understand. It sounds strange to calibrate fixed reanalysis data. I guess you mean to calibrate the initial/boundary conditions extracted from the reanalysis?

**We have modified the text (Sect. 2.4) to make it clearer:**
"To ensure that the spatial mean of the modeled SSH fits the observed SSH, the boundary conditions for the SSH from ORAS5 were bias corrected with respect to the SSH at Helgoland from the GESLAv3.0 observational data (Haigh et al., 2023). This bias was calculated on the basis of a 5-year test simulation with uncorrected boundary conditions."

Figure 2 (a) if the bias of the SST is locally up to 1.5K, please adjust the colorbar. Please also consider using discrete steps. In addition, the coloring of the contour lines are hard to follow.

**The colorbar in Fig. 2 was adjusted to 2 K. Additionally, the corresponding sentence (Sect. 3.1) was adapted to** "ROAM-NBS shows a cold bias of locally up to **2** K".
**The colorbar with discrete steps for the SST bias was tested as well. It was concluded that more details are visible with the continous color bar.**

L251: I am not exactly sure why the bias of the SST in the northern part cannot be discussed. If you mean that you cannot compare it because ROAM-NBS is overestimating the sea ice, fixing the SST whereas the OBS show different temperatures, please just mask this area.

**Thank you for this remark, it led us to re-consider the treatment of sea ice points for the SST evaluation (Sect. 3.1). The contours of the sea ice extent were removed in the final Fig. 2, since we decided to apply the time-evolving sea ice mask of the observations for calculating the SST bias between the model experiment and the observation. This is described now in the manuscript:**
"During the winter (DJF) and spring (MAM) seasons, the area in the Northern Baltic is covered by sea ice with varying extent over time. In the processed Copernicus observations, the SSTs are artificially set to -1.8°C over the regions covered by sea ice. The points where these artificial SST values are found are masked out for the calculation of the mean differences."

Figure 3: I am not sure if an average is suitable here, because you mix positive and negative anomalies. I feel that something like a RMS would be better here, but I am not convinced myself.

**The cancelling out of positive and negative biases by a spatial average is particularly true for the Baltic Sea. For the main part of the article, we decided to keep the mean biases to give an idea of their sign. Additionally, we included the RMSE time series in the Appendix (Fig. A4; the DJF and JJA bias time series were duplicates of Fig. 3 anyway) and refer to them in Sect. 3.1:**
"As the spatial averaging of biases may cancel out positive and negative values, the time series of the RMSE are additionally shown in Fig. A4. Especially in the Baltic Sea, the RMSE is, with values of about 1.5 K in summer, higher than the absolute values of the mean bias.

In the Open Atlantic and the North Sea, it is comparable to the mean bias, with about 0.75 K in all seasons."

[Figure]

**Additional Figure: RMSE time series per season and domain as shown in Fig. A4 (here including the Whole Domain).**

L263: What kind of tuning is applied. You mean expert tuning to adjust certain cloud parametrization schemes? Please name it shortly.

**Yes, it is a kind of expert tuning. The** *allow_overcast* **parameter, which was introduced by colleagues of Israel Met Service for a more direct control on cloud cover, is decreased (which increases the steepness of the distribution function of total water used to determine cloud cover when relative humidity is above a certain threshold) to increase cloud cover.**
**According to a recommendation of reviewer 1, we also moved the main part of the description to Sect. "2.1 ICON-CLM":**
"The latest tuning for ICON-CLM was done in a joint effort of the UDAG project and the CLM community (an article on the tuning of ICON-CLM for CORDEX-CMIP6 is under

preparation). For ICON-CLM as well as for ROAM-NBS, an important tuning target was the reduction of a positive surface shortwave radiation bias over land and ocean. In the current version, a tuning of the cloud cover scheme (monthly varying allow_overcast parameter) is applied. One setting that was not modified in ICON-CLM-UDAG compared to NWP, but in our ICON-CLM and ROAM-NBS simulations was the minimum diffusion coefficient for heat tkhmin. It was increased from 0.6 to 0.8 for the winter months December through February. The effects of this modified setting will be discussed in Sect. 3.2.1."

L271: What is the reason for the decrease? If not discussed, you can also delete it.
→ **deleted**

L274: Many repetitions of time series

**The whole paragraph about the SST bias time series (Sect. 3.1) was revised:**
"Time series of the spatial mean seasonal SST biases against the Copernicus observations are shown for different regions in Fig. 3. The outlines of these regions (Baltic Sea, North Sea, Open Atlantic, and the whole domain) are shown in Fig. 1. As for the bias maps, points with ice cover in the observations were masked out for the calculation of the biases. For the whole domain, the area-averaged bias is about -0.5 K for both simulations in winter (DJF, Fig. 3a, upper panel). In summer (JJA, Fig. 3a, lower panel), the bias is slightly larger for NEMO-NBS (about -0.75 K), but smaller for ROAM-NBS. However, this smaller bias for ROAM-NBS in summer is due to the higher warm bias in the Atlantic (Fig. 3b, lower panel), combined with a negative bias in the Baltic Sea (Fig. 3d). The magnitudes of the biases for the Open Atlantic region and the North Sea are similar to those for the whole domain, or even smaller. In the Baltic Sea region, the SST biases in both simulations fluctuate around zero in winter while reaching -0.75 K to -1 K during the summer season."

Figure 6: Typo at the beginning: It should be Seasonal → **done**

L362: But isn't March also the peak of the ice season?

**The misleading expression was corrected and a more complete explanation added in Sect. 3.3.1:**
"While the start of the ice growth season and the growth rate in the end of December are well captured, the sea ice extent is overestimated during the peak in the late winter (February-March) by both simulations (mean annual time series of sea ice extent not shown). Furthermore, the sea ice season is prolonged towards May compared to the observations."

[Figure]

These mean annual cycles of sea ice extent in km² (averaged over three 10-year windows, with ROAM-NBS on the left-hand side and NEMO-NBS on the right) were included in the first draft of the article. Amongst other plots, we omitted them for submission to keep a reasonable length of the manuscript.

L368-370: If the solution is obvious I am curious why it was chosen to not use the dynamical model in NEMO as it can be turned on with a simple namelist switch.

We did several test experiments to find a well working configuration. When switching on sea ice dynamics, the result was an extended sea ice area but with lower ice concentrations in the vicinity of the Bothnian Bay and Gulf of Finland, i.e. enhanced advection and spreading of the produced sea ice.
In the manuscript in the same section we state that there are lower SST and underestimated salinity during the ice season compared to observations, which favor more production of sea ice rather than melting, when advected to the open water.
We modified the text to make it clearer (Sect. 3.3.1):
"Since only thermodynamical processes are modeled within ROAM-NBS and NEMO-NBS, further enhancements in results could be obtained by also considering ice dynamics. First tests with ice dynamics did not improve the overestimation of sea ice in spring. They will need to be carried out in the future in combination with further parameter tuning and recalibration to eliminate the temperature and salinity bias in the Northern Baltic."

[Figure]

Additional figure: Sea extent, without ice dynamics on the left-hand side (top: observation, bottom: simulation), test with ice dynamics on the right.

L380: At least in Figure 10 there is no seasonal signal depicted. I think Figure 10 is too cluttered. The mean profile already gives a good impression about the mean state and the stratification. Instead of plotting SD i would appreciate a second Figure where seasonal temperature profiles are shown (JJA, DJF). Maybe this could be a new Figure A5 because as of right now it is impossible to see differences. The quality of the figure should be improved. I also think the markers just make it hard to see the curve.

Thank you for this remark. We have updated Figs. 10 and 12 to display now the mean profiles for summer and winter witout the standard deviation (see below). We have also revised the layout to improve clarity and better meet the reviewers' expectations. Correspondingly, the text in Sections 3.3.2, 3.3.3, Conclusions and A2 has been revised to reflect these new figures.
For transparency, all mean profiles and seasonal mean profiles across the four seasons are displayed below for temperature and salinity.

(b)

[Figure]

**New Fig. 10b.**

[Figure]

**New Fig. 12.**

[Figure]

Additional Figure: mean temperature profiles for all seasons.

[Figure]

Additional Figure: mean salinity profiles for all seasons.

L384: I would say the immediate layer is not there at all. If I remember correctly this is a hint that the small inflows are not captured correctly. This would also add up to the negative salinity bias.

**The comment regarding the standard deviation in line 384 (Sect. 3.3.2) has been removed from the manuscript, as the corresponding information is no longer shown in the figures. The seasonal mean temperature profiles indicate that an intermediate layer is present, although its magnitude is underestimated. It is possible that this feature was not clearly depicted in the previous figures, which may have contributed to misunderstandings. Furthermore, the seasonal analysis demonstrates that at stations without ice cover, the intermediate layer can be identified, but again appears underestimated. Corresponding remarks have been added to Section 3.3.1, including the sentence:**

"In summer, the temperature profiles also display an intermediate layer, although its magnitude is underestimated in both model runs."

L393: The evaluation is something the authors do. I would propose: The SST bias fluctuates around zero …

**As Sect. 3.3.1 was re-written largely due to the update of Fig. 10b, this sentence is deleted in the new version.**

L395: What is the reason for the shallow depth of the Landsort Deep? I get that you need to probably define a max depth, but why is it not at least the same depth at station BY15 at the Gotland Depth?

**The shallow depth of the Landsort Deep in the simulations results from two factors: the Laplacian smoothing applied to the EMODNET bathymetry and the use of the nearest grid cell for the station-based analysis. In the smoothed bathymetry, the deepest grid cell in the Landsort Deep reaches 370 m; however, this cell does not coincide with the grid point nearest to station SMHIBY31. An explanatory sentence is added in the manuscript (Sect. 3.3.2):**
"The shallow depth of the Landsort Deep in the simulations arises from Laplacian smoothing of the EMODNET bathymetry and the use of the nearest grid cell for station SMHIBY31, so that the deepest smoothed cell (370 m) does not align with the station's grid point."

L402: It is not shown how momentum fluxes compare in both simulations setups.

**With the new Fig. 10, the whole Sect. 3.3.2 was revised and the mentioned statement was removed.**

L413: What do you mean by too strong prescribed inflow of fresh water? Aren't these observations? Again time series plots of bottom salinity at different stations (BY2, BY5, BY15) could help to see that the problem is related to missing inflow intensities.

**The river runoff applied in the ocean model combines observational data with model outputs from the WaterGAP model, forming a comprehensive monthly dataset, i.e. it is not completely observation-based. In Sect. 3.3.3, we added:**
"Time series of surface and bottom salinity at stations SMHIBY2, SMHIBY5, and SMHIBY15 are shown in Figure A6, illustrating the temporal variability at each site. Both models underestimate the salinity in deeper layers but fit observations in upper layers."
**A similar statement was also added in the conclusions. The observational measurements are taken further upstream than the model input locations, which may influence the freshwater distribution in the simulations. Recent model runs utilizing the E-HYPE dataset have yielded improved results, particularly in reproducing coastal salinity patterns and bottom salinity time series at Baltic monitoring stations. The bottom salinity time series at stations SMHI BY5 and SMHI BY15 show no indication**

of missing inflow events, as the magnitude of salinity increases aligns well with observational data from NEMO-NBS. However, a persistent background bias, already present in the initial conditions of the evaluation runs, remains evident throughout the simulations.

L421: The underestimation seems like that the model is roughly 50 percent off at the bottom? Or am I mistaken, maybe I am misinterpreting the colorbars.

**To improve the understanding and scientific relevance, statistical values are calculated based on the bottom salinity time series at the three suggested Baltic stations. The bottom salinity bias is added in the manuscript text in Sect 3.3.3:**
"The surface salinity biases derived from the time series (Fig. A6) range between 0.17 and 0.37 psu for NEMO-NBS, and between 0.05 and 0.41 psu for ROAM-NBS. At the bottom, the biases are more pronounced, spanning from −1.55 psu at SMHIBY2 to −4.38 psu at SMHIBY15 for NEMO-NBS, and from −1.87 psu at SMHIBY2 to −4.64 psu at SMHIBY15 for ROAM-NBS."
**As can be seen in the normalized root mean square deviation (NRMSD) the error is not 50 percent but deviates between 18 % and 29 % for bottom salinity.**
**Results for ROAM-NBS and NEMO-NBS are:**
**SMHIBY2:**
> **Mean of observations: 15.502 psu**
> **Bias ROAM-NBS: -1.874 psu**
> **Bias NEMO-NBS: -1.553 psu**
> **RMSD ROAM-NBS: 3.134 psu**
> **RMSD NEMO-NBS: 2.838 psu**
> **Correlation coefficient ROAM-NBS: 0.488**
> **Correlation coefficient NEMO-NBS: 0.566**
> **NRMSD ROAM-NBS: 0.202**
> **NRMSD NEMO-NBS: 0.183**

**SMHIBY5:**
> **Mean of observations: 16.470 psu**
> **Bias ROAM-NBS: -4.643 psu**
> **Bias NEMO-NBS: -4.384 psu**
> **RMSD ROAM-NBS: 4.775 psu**
> **RMSD NEMO-NBS: 4.484 psu**
> **Correlation coefficient ROAM-NBS: 0.552**
> **Correlation coefficient NEMO-NBS: 0.671**
> **NRMSD ROAM-NBS: 0.290**
> **NRMSD NEMO-NBS: 0.272**

**SMHIBY15:**
> **Mean of observations: 12.571 psu**
> **Bias ROAM-NBS: -2.738 psu**
> **Bias NEMO-NBS: -2.520 psu**
> **RMSD ROAM-NBS: 2.785 psu**

**RMSD NEMO-NBS: 2.551 psu**
**Correlation coefficient ROAM-NBS: 0.434**
**Correlation coefficient NEMO-NBS: 0.672**
**NRMSD ROAM-NBS: 0.222**
**NRMSD NEMO-NBS: 0.203**

L424: I fear that this won't be enough. I think a careful investigation of the conditions at the NS BS interface driving the inflows will be necessary and potentially with subsequent recalibration of vertical and horizontal mixing. May be a slightly longer paragraph here is needed.

**We acknowledge that longer spin-ups may not fully resolve the issue; however, our current results indicate that they do lead to improvements in bottom salinity. As suggested, the sentence in L424 of the preprint has been revised and expanded into a slightly longer paragraph (can be found at the end of Sect. 3.3.3 of the revised manuscript):**
"Overall, an underestimation of bottom salinity in deeper Baltic basins can be observed. This underestimation could be mitigated in future simulations by employing longer spin-up periods or improved initial conditions. Additionally, efforts should focus on refining vertical and horizontal mixing within the Baltic and increasing spatial resolution in the Danish Straits. For climate projections based on the current setup, bias correction of bottom salinity values in basins deeper than 40m is necessary."

Figure 12: I think the figure hints that the inflow dynamics are probably not really well resolved. Again time series plots would help to clarify this. Also the stratifications seem to be underestimated significantly. For future studies processes such as oxygen transportation into the Baltic Sea won't be correctly estimated.

**As suggested, the salinity time series plots have been included in the appendix. These plots show that, while the magnitudes of the major and minor inflows are slightly underestimated in the time series of the lower levels, the overall dynamics are well captured. We acknowledge that further improvement in stratification will be necessary for future coupling with biogeochemical models, and a corresponding note has been added to Sect. 3.3.3:**
"Time series of surface and bottom salinity at stations SMHIBY2, SMHIBY5, and SMHIBY15 are shown in Figure A6, illustrating the temporal variability at each site. Both models underestimate the salinity in deeper layers but fit observations in upper layers."

L430: Why only a short period?

**Following explanation was added to section 3.3.4.:**
"A period was selected during which all observational stations provided continuous hourly data, ensuring that the resulting statistics are fully comparable."

Figure 14: I think the colorbar makes it hard to see the differences. At a first glance it

looks like that the inflow is not reaching the deeper basins. Would anomalies help here or a different coloring? Maybe you have a better idea.

**Thank you for this remark. By applying discrete colorbars and removing the contour lines, we hope that Figures 14 and 15 are now easier to interpret.**
**It is indeed correct that the inflows do not reach the deepest basins, as also described in lines 477– 482 of the preprint. Nevertheless, the model setup is considered as production-ready for the EURO-CORDEX domain, provided that a post-simulation bias correction for bottom salinity and temperature within the Baltic Sea is applied. The summarizing sentence in Sect. 3.3.3 was changed to**
"Overall, an underestimation of bottom salinity can be observed and shall be taken into account for bias-correction of future climate projections."

L582: Freshwater is from observations? How can it be too strong?

**As mentioned above and in the text, the runoff data is a combination of observational and model data. Also, within the current NEMO-NBS and ROAM-NBS setups, the runoff is introduced in the upper layer and no enhanced treatment to feed it into the model throughout the water column, available in NEMO, is applied. Two entences discussing these options were added in the Conclusions (also mentioning the coupling of a runoff model as asked by reviewer 1):**
"These [salinity] biases may be attributed to overly strong prescribed freshwater runoff, which is not exclusively observation-based, and insufficient representation of saline inflow from the North Sea. Prescribing runoff throughout the water column and enhancing vertical mixing could improve sea surface salinity in NEMO-NBS and ROAM-NBS simulations. For the generation of the historical simulations, an online coupled runoff model will be used as in Hagemann et al. (2024), which is already available in the setup but was not used for better comparability between the coupled and the ocean-only simulation."

L585: I am not sure if the stratification is reproduced as well as the inflows.

**That's correct. The stratification in the Baltic Sea is underestimated from the outset of the simulations, as is particularly evident in the salinity time series. Improving the representation of stratification could be achieved through enhanced initial conditions or by fine-tuning the vertical mixing parameterizations. The sentence in the conclusion was adjusted to:**
"The major inflow events are qualitatively reproduced but quantitatively underestimated."
**Within the following section in the Conclusions, the limitations of the modeled stratification are discussed for the Baltic (lines 587-597 in pre-print).**

Overall, this is a valuable and well-structured contribution, and I appreciate the effort that went into both the coupled setup and the detailed evaluation. Addressing the points above will further strengthen the scientific robustness and clarity of the study.